# Inflammatory stromal and T cells mediate human bone marrow niche remodeling in clonal hematopoiesis and myelodysplasia

Karin D. Prummel [1,20], Kevin Woods[2,3,20], Maksim Kholmatov [1,4,5,20], Eric C. Schmitt [2,3,6], Evi P. Vlachou [1], Mayssa Labyadh[7,8], Rebekka Wehner[9,10,11], Gereon Poschmann [12], Kai Stühler[12,13], Susann Winter [14], Uta Oelschlaegel[14], Manja Wobus [11,14], Logan S. Schwartz[15], Pedro L. Moura [16], Eva Hellström-Lindberg [16], Krishnaraj Rajalingam [17], Matthias Theobald[2,3], Jennifer J. Trowbridge [15], Clémence Carron [7,8], Thierry Jaffredo[7,8], Marc Schmitz[9,10,11], Uwe Platzbecker[18], Judith B. Zaugg [1,5] ✉ & Borhane Guezguez [2,3,19] ✉

Somatic mutations in hematopoietic stem/progenitor cells (HSPCs) can lead to clonal hematopoiesis of indeterminate potential (CHIP) and progression to myelodysplastic syndromes (MDS). Using single-cell and anatomical profiling of a large cohort of human bone marrow (BM), we show that the HSPC BM niche in CHIP and MDS is undergoing inflammatory remodeling. This includes loss of CXCL12$^+$ adipogenic stromal cells and the emergence of a distinct population of inflammatory mesenchymal stromal cells (iMSCs), which arise in CHIP and become more prevalent in MDS. Functional studies in primary BM HSPC-MSC co-cultures reveals that healthy aged and CHIP HSPCs activate stromal support, while MDS HSPCs fail to do so. In contrast, MDS blasts further suppress HSPC support and trigger inflammation, indicating disease-stage-specific stromal disruption. In parallel, we show that iMSCs retain partial support and angiogenic potential in MDS, coinciding with expanded BM vasculature. Additionally, we identify IFN-responsive T cells that preferentially interact with iMSCs, potentially reinforcing local inflammation. These findings position iMSCs as central mediators of early BM niche dysfunction and potential therapeutic targets for intercepting pre-malignant hematopoiesis.

Aging is associated with a decline in hematopoietic system function and alterations in the composition of the bone marrow (BM) stem cell niche. Furthermore, it increases the risk for hematopoietic stem and progenitor cells (HSPC) to accumulate somatic mutations in cancer-driver genes[1]. Some of these mutations confer a competitive advantage and drive clonal expansion[2]. This phenomenon, when the variant allele frequency (VAF) exceeds 2% in the absence of hematological neoplasms, is termed clonal hematopoiesis with indeterminate potential (CHIP)[3,4] and is detected in more than 10% of adults over 65[5,6].

Most recurrent CHIP mutations affect DNA methylation regulators (*DNMT3A, TET2, ASXL1*) and have been associated through a complex array of confounding factors with increased risks of non-malignant disorders such as cardiovascular disease and rise in overall mortality[7–10]. Importantly, CHIP may precede and increase the risk of developing hematologic malignancies such as myelodysplastic syndromes (MDS) and transformation to acute myeloid leukemia (AML)[11–13]. As CHIP is increasingly recognized as a biomarker for the risk of blood malignancy development, understanding its impact on

the BM microenvironment may provide a better insight into the mechanisms driving disease progression.

MDS represents a group of hematological malignancies arising from mutated HSPCs, resulting in ineffective hematopoiesis, BM dysplasia, peripheral blood cytopenia, and predisposition to secondary AML[14]. While MDS shares some genetic features with CHIP, such as mutations in epigenetic regulators, it also exhibits cytogenetic aberrations and additional mutations, especially in RNA splicing factors (*SF3B1*, *SRSF2*, *ZRSR2*)[15,16]. For instance, *SF3B1* mutations are highly predictive for the occurrence of anemia and diagnosis of myeloid neoplasm[17]. In particular *SF3B1* mutations are found in 80% of low-risk MDS cases characterized with ring sideroblasts (RS), thereby defining a distinct MDS-*SF3B1* subtype within MDS-RS[18–20].

The BM microenvironment (BM niche), including immune cells, endothelium, and mesenchymal stromal cells (MSC), constitutes specialized niches that orchestrate HSPC differentiation and maintenance, bone homeostasis, and immunity[21,22]. These niches undergo remodeling with aging, marked by a shift of MSC differentiation towards the adipogenic lineage and increased adipocytes (fatty BM)[23–25]. Another hallmark of BM aging is the emergence of inflammaging, a sterile low-grade inflammatory state characterized by elevated levels of pro-inflammatory factors including IL-1β, IL-6, and TNFα[26–28]. Inflammaging is associated with a decreased stem cell function[29,30] and has been observed across multiple organ systems[31,32]. In CHIP carriers, elevated blood serum levels of IL-6, CXCL8, IL-18, and TNFα have been detected compared to age-matched controls[33–35]. Furthermore, the establishment of an inflammatory BM state has been shown to be beneficial for clonal expansion of *TET2*- and *DNMT3A*-mutant HSPCs in mouse models as well as in human CHIP cohort studies[30,35–38].

Extensive remodeling of the BM niche has been observed in hematopoietic malignancies, fostering an environment that supports malignant cells. This remodeling particularly influences the stromal components, which can alter the functions of resident immune cells, such as T cell-mediated responses[39–41]. In MDS, in vitro cultured MDS-derived BM MSCs show an inflammatory program[42], decreased proliferation capacity, and reduced HSPC support[43] compared to healthy donor-derived BM MSCs. A recent study in a *DNMT3A*-mutant CHIP mouse model (equivalent to a human VAF of 1–4%) suggests that mutated HSPCs can induce senescence in BM MSCs through pro-inflammatory cytokines including IL-6 and TNFα[44]. However, in vivo studies on the MDS BM niche cells remain scarce, and the extent to which CHIP-mutant clones can remodel the BM microenvironment remains unclear, particularly in the human context and at single-cell level.

In this study, we investigate the BM microenvironment in aged controls, CHIP carriers (*DNMT3A*, *TET2*), and low-risk MDS patients (*DNMT3A*, *TET2*, spliceosome mutations) using multimodal approaches, including targeted RNA profiling, flow cytometry, single-cell sequencing, and imaging. Our analyses reveal distinct alterations in stromal and immune compartments, marked by the emergence of an inflammatory MSC subset in CHIP that further expands in MDS. We show that these MSCs act alongside IFN-responsive T cells to remodel the niche and interact with mutated HSPCs in ways that may sustain malignant hematopoiesis. Together, these findings contribute to a deeper understanding of the inflammatory human BM microenvironment in CHIP and MDS and open avenues for therapeutic intervention.

## Results

### Distinct inflammatory and immune microenvironment profiles in CHIP and MDS bone marrow

To understand how the microenvironment is altered across CHIP and MDS, we analyzed bone marrow (BM) samples from a balanced cohort of 84 donors, including 35 age-matched Controls, 17 CHIP donors (VAF ≥ 2%), and 32 MDS patients, the majority of whom had low-risk disease and were newly diagnosed or disease-modifying treatment-naïve cases (i.e., erythropoietin or supportive care only) (Fig. 1A, and Table 1). *DNMT3A* and *TET2* were the most frequently mutated genes across most CHIP and MDS donors. In addition, the majority of MDS patients also harbored at least one mutation in a spliceosomal gene, such as *SF3B1* (*n* = 16), *SRSF2*, or *U2AF1* (*n* = 6) or carry only *DNMT3A*/*TET2* mutations (*n* = 10) (Table 1, Supplementary Data 1, Supplementary Fig. 1A). Low-frequency mutations (VAF < 2%) identified in a few controls showed no association with clinical features (including CRP values), supporting their classification as non-CHIP in our BM cohort (Supplementary Data 1, Supplementary Fig. 1A). Blood count analysis showed equal proportions of leukocytes across the mutational spectrum of the MDS cohort but indicated increased ring sideroblasts (RS) and anemia specifically in *SF3B1*-mutated MDS (Supplementary Fig. 1B), which is consistent with the current WHO consensus classification of MDS-RS[45,46].

To gain a comprehensive overview of microenvironmental alterations, we profiled bulk gene expression in total BM mononuclear cells (MNC) using two NanoString nCounter panels targeting 773 immune-related (Immune Exhaustion) and 730 cancer inflammation-associated (PanCancer Immune Profiling) genes respectively (Supplementary Data 2, and Fig. 1B). Principal component analysis (PCA) revealed a separation of MDS samples from CHIP and Control along the main principal component in both panels, with only one MDS outlier (Fig. 1C; **Methods**). Correlation heatmaps of the top 10 principal components (PC1–10) with clinical metadata showed no strong associations, apart from the expected alignment with VAF, indicating that the transcriptional patterns were largely independent of other clinical parameters (Supplementary Fig. 2A). Although cluster separation appeared slightly more distinct in the Immune Exhaustion panel, both panels consistently captured transcriptional divergence in MDS, reflecting broad remodeling of the BM immune milieu.

Differential gene expression analysis identified 123 genes significantly altered between MDS and Controls (FDR < 0.05), with the majority being downregulated in MDS BM (Supplementary Fig. 2B). Gene set enrichment analysis (GSEA) revealed upregulation of TNFα and IFNα pathways in MDS, indicating a pro-inflammatory environment, while T cell-related processes were downregulated (Fig. 1D).

To further identify which cell types contributed to the transcriptional changes in the MDS niche, we mapped the differentially expressed genes onto a recently published single-cell atlas of healthy BM[47]. For each gene, we computed average expression across annotated cell types and clustered the resulting gene-by-cell-type matrix (Supplementary Fig. 2C), followed by grouping cell types with similar patterns into the major BM cell populations (Fig. 1E). This revealed that genes downregulated in MDS are predominantly expressed by lymphoid populations, including T cells, pre/pro-B cells, mature B cells, and B plasma cells (e.g., *CD3*, *CD8*, *CD4*, *CD22*, *IGH*, *CD38*, *CD19*) (Fig. 1E). This overall loss of B cells was confirmed by immunophenotyping of the BM (Fig. 1F, Supplementary Fig. 1C) and is consistent with prior studies on low-risk (LR) MDS and *SF3B1*-mutated MDS[48–51]. In contrast, the total numbers of CD3⁺ T cells, CD4⁺/CD8⁺ subsets, NK cells, NKT cells, and granulocytes were unchanged across BM conditions (Supplementary Fig. 1C, D). Together with the upregulation of several pro-inflammatory genes in MDS BM (e.g., *TNF*, *IFNL1*, *CCR9*) (Fig. 1E), these findings suggest the presence of a phenotypically preserved but functionally altered T cell population contributing to the inflammatory milieu in MDS.

In addition to lymphoid alterations, MDS donors exhibited transcriptional shifts across other BM compartments, including genes specific to early myeloid precursor cells (Fig. 1E), likely reflected by the elevated presence of MDS myeloblasts (CD34⁺/CD117⁺) as confirmed by immunophenotyping (Fig. 1F). Moreover, dendritic cell–specific genes were reduced, whereas neutrophil- and myeloid progenitor–associated transcripts were upregulated, suggesting changes in abundance or functional state of these populations (Fig. 1E).

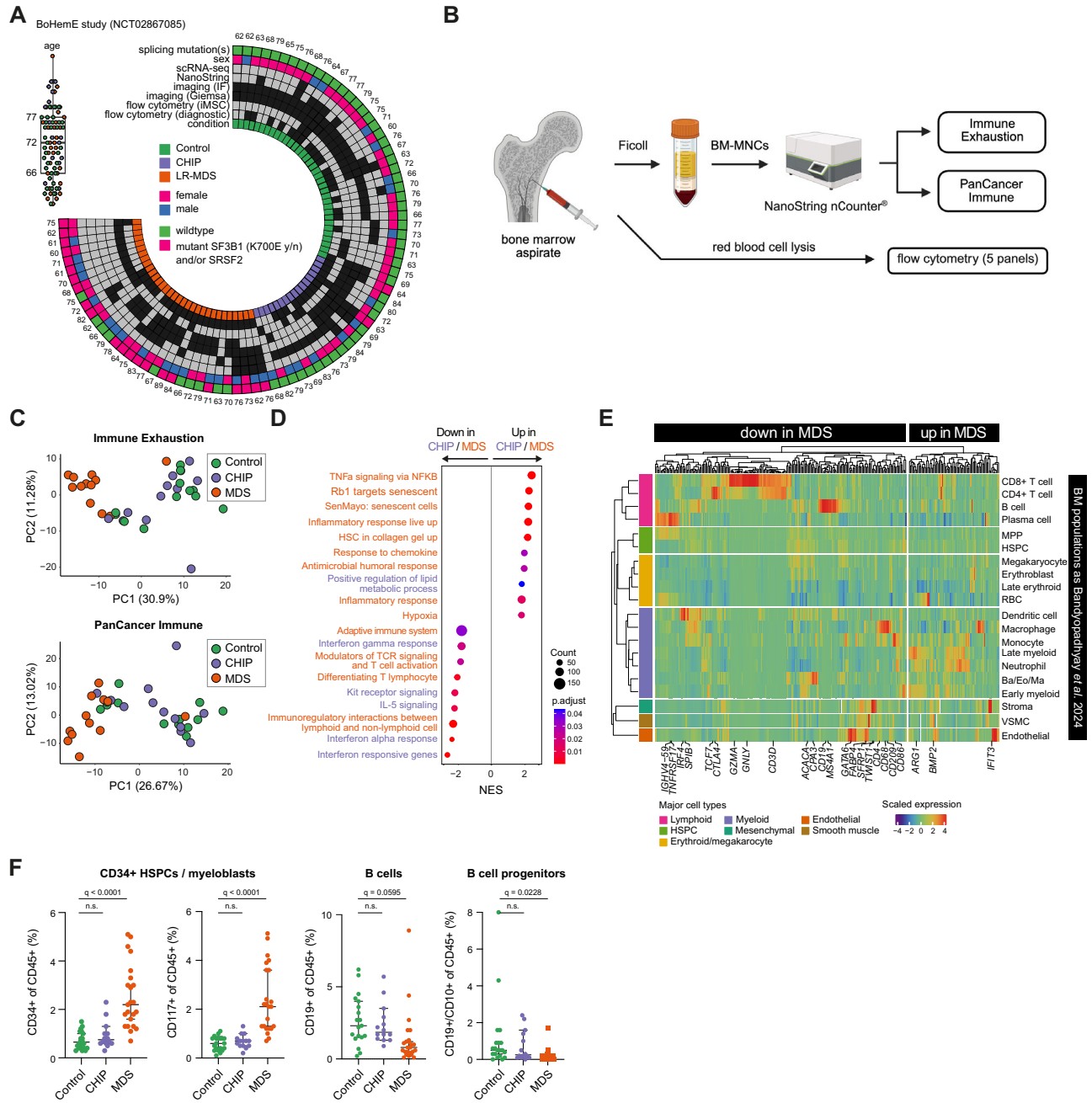

**Fig. 1 | Inflammatory expression profiling in bulk bone marrow across CHIP and MDS. A** Circular heat map of the BoHemE cohort (NCT02867085), showing age distribution (boxplot, top left), sex (female: magenta, male: blue), splicing mutations (*SF3B1*, *SRSF2*; wildtype: green, mutant: magenta), and conditions (green: Control, purple: CHIP, orange: MDS). Rings indicate inclusion (black boxes) in various analyses such as scRNA-seq, NanoString, flow cytometry, and immunofluorescence analysis. **B** Schematic of the experimental workflow. Bone marrow (BM) aspirates were processed into BM mononucleocytes (BM-MNCs) using Ficoll separation or only erythrocyte lysis. Ficoll-treated samples were used for NanoString profiling (Immune Exhaustion and PanCancer Immune panels) and erythrocyte-lysed samples for flow cytometry analysis to quantify major cell types (see Supplementary Data 2 for NanoString panels and Supplementary Data 3 for flow cytometry panels). Created in BioRender. Prummel, K. (https://BioRender. com/pzpb028). **C** Principal component analysis (PCA) scatter plots showing the separation of Control, CHIP, and MDS samples (*n* = 12/condition) by the

Immune Exhaustion (top) and PanCancer Immune (bottom) panels. Each dot represents a sample, colored by condition (green: Control, purple: CHIP, orange: MDS). **D** Gene Set Enrichment Analysis (GSEA) results illustrating gene sets significantly up- or downregulated in MDS vs. Control (orange) and CHIP vs. Control (purple). *P* values were adjusted using Benjamini–Hochberg procedure. **E** Heatmap of NanoString-derived genes differentially expressed between MDS and Control samples (from **D**) onto scRNA-seq data from healthy BM[47]. Genes are averaged by cell type and Z-scored across cell types for visualization. **F** Flow cytometry analysis comparing the relative proportions of CD34$^+$/CD117$^+$ HSPCs/myeloblasts, CD19$^+$ B cells, and CD10$^+$CD19$^+$ B cell progenitors within the CD45$^+$ cell population across Control (*n* = 20), CHIP (*n* = 14), and MDS (*n* = 24) donors. Medians with 95% confidence intervals are shown. Statistical test: one-way ANOVA with FDR for multiple comparison correction. Source data are provided in the Source Data file. VSMC vascular smooth muscle cell, n.s. not significant.

Lastly, stromal gene expression revealed reduced levels of *TWIST* and *SFRP1* (markers involved in MSC identity and differentiation) while endothelial-associated genes (*BMP2, IFIT3*) were upregulated in MDS (Fig. 1E). Since these genes are important modulators of HSPC and osteogenesis[52–55], our data indicate that both stromal and endothelial compartments might also undergo significant remodeling in MDS BM.

Although CHIP donors did not show statistically significant expression changes compared to Controls (Supplementary Fig. 2D), B and T cell-related gene sets were upregulated relative to MDS (Supplementary Fig. 2E), indicating preserved lymphoid integrity. Notably, stratification by clone size further revealed that CHIP donors with VAF ≥ 5%, primarily carrying *DNMT3A* mutations clustered more closely to MDS in the PCA (Supplementary Fig. 2F) and showed increased expression of inflammatory and proliferative markers compared to CHIP/Control donors with VAF < 5% (Supplementary Fig. 2G). This suggests that a subset of CHIP donors may already harbor an inflammatory BM microenvironment. In contrast, Controls with low-frequency mutations (VAF < 2%) showed no association with molecular features (Supplementary Fig. 1A, and Supplementary Data 1), underscoring their distinction from CHIP donors.

Overall, the transcriptional analysis of bulk BM reveals a pro-inflammatory microenvironment that emerges in high-VAF CHIP and becomes more pronounced in MDS, accompanied by distinct changes in cellular composition. These include expansion of immature myeloid cells, selective loss of B cells, and remodeling of the T cell, stromal, and endothelial compartments, suggesting stepwise BM niche disruption across distinct disease states.

### Single-cell RNA-seq reveals inflammatory stromal and lymphocyte subsets in the BM niche of CHIP and MDS

To further investigate the observed remodeling of the BM microenvironment in CHIP and MDS, we employed single cell transcriptomics on selected BM populations from a representative subset of the total cohort, comprising 3 Control, 3 CHIP (*DNMT3A* and/or *TET2*), and 4 MDS patients (*SF3B1/SRSF2* and *DNMT3A* and/or *TET2*) (Fig. 2A, Supplementary Fig. 1A; more info in Supplementary Data 5). We focused on three key compartments of the BM: non-hematopoietic stromal fraction (negative gating strategy: CD45⁻CD235a⁻CD71⁻CD14⁻CD38⁻), HSPCs (CD45⁺CD34⁺CD235a⁻CD71⁻CD14⁻), and T cells (CD45⁺CD3⁺), while minimizing the contamination by erythroid and other immature cells including MDS-RS (Supplementary Fig. 3A). To enhance stromal cell recovery, we additionally sorted CD271⁺ cells for plate-based scRNA-seq (CEL-Seq2). CD271, a well-established marker of human BM stromal cells[56,57], labeled the predominant stromal population in our selected BM cohort compared to other known stromal markers (Supplementary Fig. 3B–E).

After donor demultiplexing based on genotype deconvolution (**Methods**), quality control, and cell calling (Supplementary Fig. 4A–E), we identified a population of hematopoietic cells in the stromal gate despite the removal of CD38⁺/CD71⁺/CD235a⁺ cells (Supplementary Fig. 4F; **Methods**). These cells resembled monoblasts and erythroblasts (or RS) and predominantly originated from MDS donors; and were therefore removed for the main analyses. Altogether, these filtering steps yielded in total 24,693 cells, with between 1,190 and 4,393 high-quality cells per donor (median 3,644 UMIs, 1,610 unique genes, and 6.2% mitochondrial reads per cell; Supplementary Data 4; Supplementary Fig. 4A). A UMAP representation followed by broad cell type classification revealed the expected 3 major compartments: 1,396 stromal cells (*CXCL12* and *LEPR*), 9,962 T cells (*CD3E* and *CD247*), and 12,257 HSPCs (*CD34* and *SPINK2*) (Fig. 2B), along with a small population of *CD14*-low monocytes, which was unexpectedly recovered from MDS donors despite CD14⁻ gating.

To further refine the cell populations, we integrated the scRNA-seq data for each major cell population across all donors using Seurat v4 (**Methods**) and generated a UMAP embedding for each broad cell

### Table 1 | Clinical characteristics of experimental cohort

| Variable | Control | CHIP | MDS |
|---|---|---|---|
| **Total number of donors** | 35 | 17 | 32 |
| **Sex** | | | |
| Female | 27 | 10 | 15 |
| Male | 8 | 7 | 17 |
| **Age (years)** | | | |
| Median | 71 | 77 | 72 |
| Range | 60–79 | 63–84 | 60–89 |
| **Mutations (number of donors with most representative % VAF)*** | | | |
| Epigenetic regulators (DNMT3A, TET2, ASXL1) | N/A | 14 | 10 |
| SF3B1 (K700E and others) | N/A | 1 | 16 |
| Other splicing factors | N/A | 0 | 6 |
| Other mutations (PPMD1, SRP72) | N/A | 2 | 0 |
| **IPSS-R (number of donors)** | | | |
| IPSS-R ≤ 3.5 (low) | N/A | N/A | 17 |
| IPSS-R > 3.5 - 4.5 (intermediate) | N/A | N/A | 10 |
| IPSS-R > 4.5 (high) | N/A | N/A | 5 |
| **MDS Risk group (number of donors)** | | | |
| Low-Risk (LR-MDS) | N/A | N/A | 27 |
| High-Risk (HR-MDS) | N/A | N/A | 5 |
| **Treatment status** | | | |
| Newly diagnosed (no treatment) | N/A | N/A | 10 |
| Treatment-naïve cases (i.e., erythropoietin or supportive care only) | N/A | N/A | 18 |
| Hypomethylating agent (AZA) | N/A | N/A | 3 |
| Lenalidomide | N/A | N/A | 1 |

*VAF variant allele frequency; only somatic mutations with VAF ≥ 2% in defined genes are considered in this study. See details in Supplementary Data 1 and Supplementary Fig. 1

type (T cells, stroma, HSPCs including CD14-low monocytes). We next performed Leiden clustering and annotated the populations using canonical marker genes, reference human T cells and HSPC atlases (**Methods**[58–65]), and gene regulatory network inference (SCENIC[66]) to identify key transcription factor regulons (**Methods**[60–63,66]). In total, we defined 13 HSPC clusters, 9 T cell clusters, and 6 stromal cell clusters (Fig. 2C–H, and Supplementary Fig. 4G–J). For the HSPC compartment, we identified HSC and multipotent progenitors (HSC/MPP; *HLF, AVP, GPR56, MECOM*), along with lymphoid-primed multipotent progenitors (LMPP; *SPINK2, CD34, PROM1*). B lymphoid progenitors comprised both Pro-B cells, (*CD96, DNTT, EBF1*) and Pre-B cells (*BACH2, PAX5*). Myeloid commitment was represented by granulocyte-monocyte progenitors (GMP; *PRTN3, MPO*) and plasmacytoid-dendritic cell progenitors (pDC-Pr; *IRF8, CUX2*). In addition, the eosinophil−basophil−mast cell progenitor population (EBMP; *ENPP3, IKZF2*) was resolved alongside, megakaryocyte-erythroid progenitors (MEP; *GATA2*), megakaryocyte progenitors (MkPr; *PF4, PBX1*), and erythroid progenitors (EryPr; *APOC1, KLF1*). Finally, we identified a CD14-low myeloid population (*CD68, MFAB, SPI1*) that was reminiscent of non-classical CD14-low monocytes. These were previously defined as patrolling monocytes involved in the innate local surveillance of tissues and shown to expand in blood malignancies[67–69]. Two low-quality (high_mito) or donor-specific clusters were excluded from further interpretation (Fig. 2C, D).

We annotated 9 T cell populations using a combination of marker genes and regulon activities of specific transcription factors, including those inferred from SCENIC and from our previously established T cell-

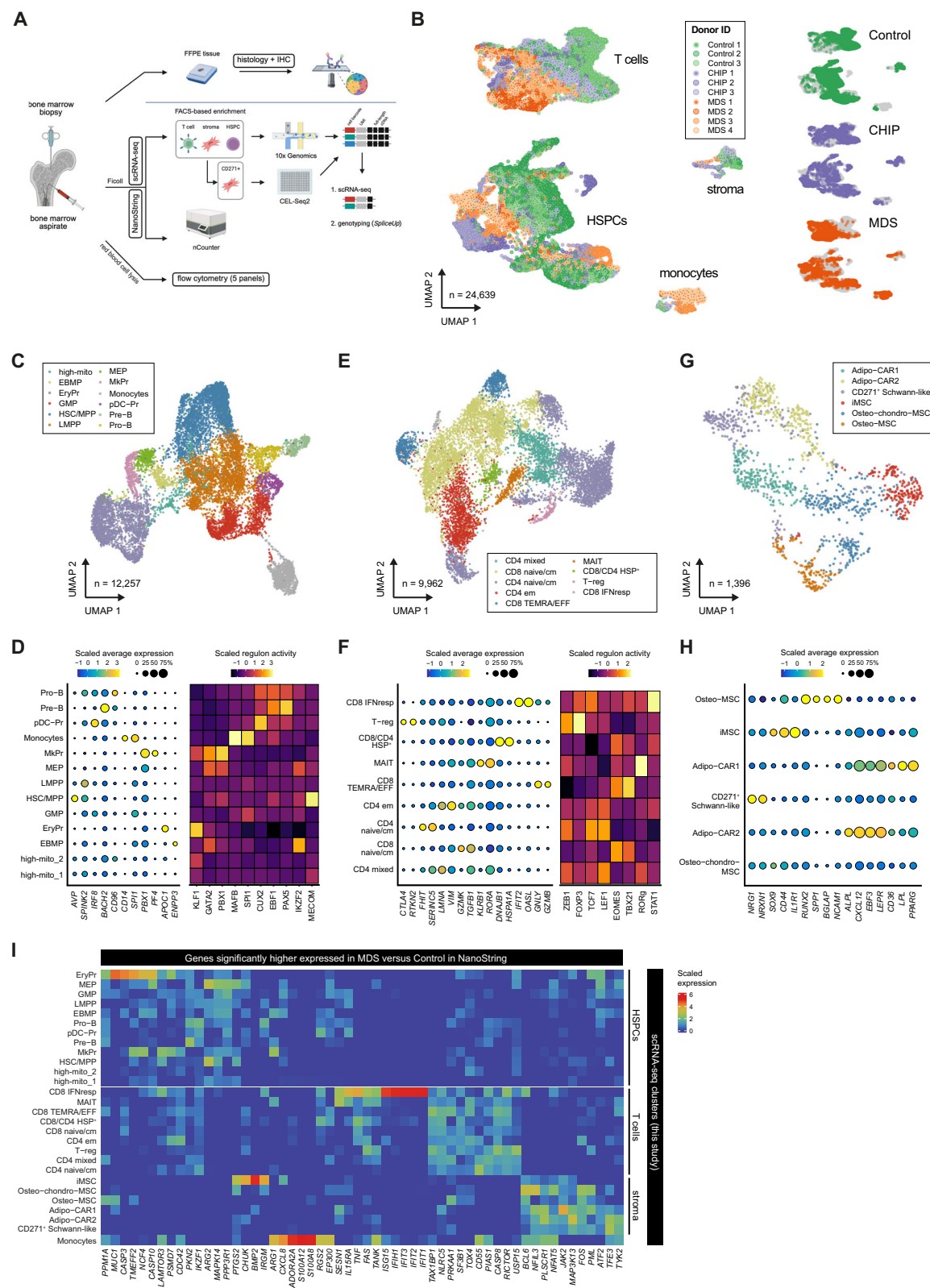

specific regulons[63,65,70,71]. The CD4 clusters included: CD4 naive/central memory (CD4_naive/CM; *SELL, CCR7*), CD4 effector memory (CD4_EM; *IL7R, KLRB1, AQP3*), a mixed CD4 population, and regulatory T cells (Treg; *IL2RA, RTKN2*, FOXP3 regulon activity). Among CD8 T cells, we distinguished CD8 naive/central memory (CD8_naive/CM; *CCL5, GZMK*), CD8 terminally differentiated effector memory re-expressing *CD45RA* (T_EMRA; *FCGR3A, GZMB, GNLY*), as well as IFN-responsive CD8 effector

cells (CD8 IFN-responsive; *OASL, IFIT2*). Additionally, we observed a mixed CD4/CD8 heat-shock protein-expressing population (CD4/CD8 HSP+; *DNAJB1, HSP90AA1, HSPA1A*) and mucosal-associated invariant T cells (MAIT; *SLC4A10, ME1*) (Fig. 2E,F).

Within the stromal compartment, we annotated 6 distinct subpopulations based on a BM reference atlas[47], including two adipogenic CXCL12 abundant reticular (Adipo-CAR1/2; sharing *CXCL12, LEPR*, and

**Fig. 2 | Identification of BM cell subsets in Control, CHIP, and MDS by scRNA-seq. A** Schematic of the experimental design using donor-derived BM aspirates and FFPE trephine bone biopsies. Created in BioRender. Prummel, K. (https://BioRender.com/kdi1w0u). **B** Uniform manifold approximation and projection (UMAP) of 24,639 cells across the major cell populations of scRNA-seq from donors highlighted in Fig. 1A and Fig. 2A, colored by Donor ID and split by condition. **C** UMAP of 12,557 HSPCs, with clusters annotated by cell type. **D** Dot plot of marker gene expression in HSPCs. Circle size corresponds to percent of cells expressing the given gene and color the expression level. The heatmap shows average AUCell scores of marker transcription factors based on regulons inferred in this study. **E** UMAP of 9,962 T cells, with clusters annotated and colored by cell type. **F** Dot plot of T cell marker expression and heatmap of average AUCell scores[65]. **G** UMAP of 1,396 stromal cells integrated between 10x and CEL-Seq2 data, colored by cell type annotation. **H** Dot plot of stromal cell markers. **I** Average expression of genes upregulated in MDS vs. Control in the NanoString data, shown across our scRNA-seq cell type clusters. Source data are provided in the Source Data file and Supplementary Data 7.

*EBF3*, and distinguished by *PPARG and LPL* -Adipo-CAR1 and *ALPL* and *PDGFRB* - Adipo-CAR2), chondrogenic- (*SOX9*), and osteogenic-lineage (*RUNX2, SPP1, NCAM1, BGLAP*) MSCs, as well as neural crest-derived Schwann-like cells (*NRG1, NRXN1*), which express *CD271 (NGFR)* akin to MSCs[72,73] and were enriched among our CD271+ population (Fig. 2G, H). Notably, we identified a stromal population expressing inflammation-associated genes, including *CD44* and *IL1R1*[74,75], which we refer to as inflammatory MSCs (iMSCs). Other stromal cells, including endothelial cells, vascular smooth muscle cells/pericytes, or mature osteocytes and adipocytes could not be recovered from BM aspirates, either due to their low abundance or mechanical fragility during cell isolation.

The differentially expressed genes from the bulk BM transcriptomics analysis in the larger cohort were highly expressed in MDS-specific cell populations (iMSCs, IFN-responsive T cells, and non-classical CD14-low monocytes), which did not exist in the healthy BM reference, confirming that we profiled the relevant cell populations (Fig. 2I).

Overall, our single-cell data reveals distinct, disease-associated subpopulations within the stromal, T cell, and HSPCs across the CHIP and MDS BM. Integration with the bulk transcriptomics data of the larger cohort highlights the emergence of inflammatory populations in MDS, particularly iMSCs and IFN-responsive T cells, underscoring their potential role in BM niche remodeling.

## Inflammatory MSCs are exclusively present in CHIP and hematologic malignancies

To further investigate the stromal cell populations, we quantified the subtype changes across Control, CHIP, and MDS samples. The most notable shift was the emergence of the iMSC population, present exclusively in CHIP and MDS (Fig. 3A, Supplementary Fig. 4H, Supplementary Data 4). Although bulk BM from CHIP donors lacked a significant pro-inflammatory signature (Supplementary Fig. 2D), our scRNA-seq approach sensitively captured these rare iMSCs, highlighting its ability to resolve subtle and low-abundance cellular changes. This iMSC population was detected in 2 out of 3 CHIP donors and all MDS donors examined, albeit in varying proportions (Supplementary Data 4). In addition to the expansion of iMSCs, MDS patients exhibited a marked depletion of both Adipo-CAR1 and Adipo-CAR2 MSCs, while CHIP donors showed a specific loss of Adipo-CAR1. This is corroborated by the reduced expression of adipocyte-specific markers (*ACACA, FABP4*) in the NanoString MDS data (Fig. 1E), and by histological analysis, showing a significant reduction of BM adipocytes in MDS (Fig. 3B, C, and Supplementary Fig. 5A–C). These findings are in accordance with previous reports linking adipocyte loss to ineffective hematopoiesis in LR-MDS[76–78].

Given their striking enrichment in CHIP and MDS, we next characterized the transcriptional features of the iMSC population. They displayed high expression of inflammatory mediators (*CD51, CD44, IL1R1, PTGS2* (a prostaglandin synthase involved in inflammatory responses), *CXCL8, CCL2*) as well as genes involved in pro-inflammatory extracellular matrix (ECM) remodeling (*ADAM12, FN1, LAMC1, ITGA5, ITGB1, ITGB3*) and profibrotic collagens (*COL4A1, COL4A2, COL6A2, COL6A*3) (Fig. 3D). Additionally, iMSCs expressed several genes that were also significantly upregulated in bulk BM, including *CHUK* (encoding IKK-alpha, crucial for NF-κB signaling

activation and regulation), *BMP2*, and *PTGS2* (Fig. 3E)[79,80], further linking iMSCs as active players of structural remodeling of the niche.

The presence of iMSCs was confirmed by in situ immunofluorescence staining for IL-1R1 in FFPE BM tissue, revealing higher IL-1R1+ cell counts in MDS compared to Control and CHIP donors (Fig. 3F, G). Despite the overall low IL-1R1+ cell counts in CHIP, discrete clusters of IL-1R1+/CD271+ cells were still detectable in CHIP BM sections (Supplementary Fig. 6A). Additionally, flow cytometry on a subset of our full cohort of BM aspirates (Fig. 1A, Supplementary Fig. 1A; n = 22) further validated the presence of iMSCs in CHIP and MDS using CD44 and CD51/61 markers. In particular, we observed a significant increase of CD44+ iMSC within the CD271+/CD73+ stromal compartment in MDS, harboring spliceosome mutations (Fig. 3H, I, and Supplementary Fig. 6B–D). CD44 is linked to stromal inflammation and ECM production through TGFβ and RhoA-YAP signaling[81,82].

Inflammatory alterations in stromal cells have been reported in other BM malignancies such as AML[83] and mature B cell-derived Multiple Myeloma (MM)[74], where they are linked to cancer-associated fibroblast (CAF)-like signatures[84]. Using a MSC-inflammatory signature derived from these prior studies (21 genes, Supplementary Data 5, **Methods**), we confirmed MSCs from our Controls lacked inflammatory features (Fig. 3J), while CHIP/MDS-specific iMSCs showed high inflammation scores (Fig. 3J, K).

To further quantify the transcriptional similarities of our iMSCs to those previously described in other BM malignancies, we integrated our dataset with published scRNA-seq data from BM stromal compartments of healthy, MM, and *NPM1*-mutant AML patients (**Methods**[47,74,83]). This revealed that the iMSC cluster shared between CHIP and MDS represents a subset of the iMSCs found in AML and MM (Fig. 3L, M, and Supplementary Fig. 7A). This shared iMSC cluster across multiple BM malignancies exhibited a high inflammatory signature and was nearly absent in healthy BM[47] and disease-free controls from the MM and AML datasets[74,83] (Fig. 3L, and Supplementary Fig. 7A). These findings suggest a slightly increased inflammatory niche in CHIP and a more pronounced expansion of an iMSC subset in MDS, which partially resembles other BM malignancies, such as AML and MM. This suggests the existence of a shared stromal response to (chronic) inflammatory stimuli across diverse BM diseases.

## iMSCs differ between CHIP and MDS in their HSPC-support signatures

To explore how stromal remodeling and inflammation affect the HSPC support, we curated a literature-based gene signature of known HSPC-support factors (46 genes, Supplementary Data 5)[85] and assessed its expression across the stromal populations. We observed that the adipo-MSCs (Adipo-CAR1 + 2) showed the highest HSPC support score (Fig. 4A). Given the loss of these populations in MDS, and to a lesser extent in CHIP (Fig. 3A, Supplementary Fig. 7B), indicates a reduced capacity of the stromal compartment to maintain HSPCs in CHIP, with this impairment being even more pronounced in MDS. Indeed, quantifying the in situ CXCL12 protein levels, an essential HSPC support and maintenance factor[86], relative to the number of CD271+ stromal cells in BM biopsies from our cohort, revealed a significantly lower CXCL12/CD271 ratio in MDS compared to CHIP and Control (Fig. 4B, C). This suggests that either MSCs in MDS express lower levels of CXCL12, or

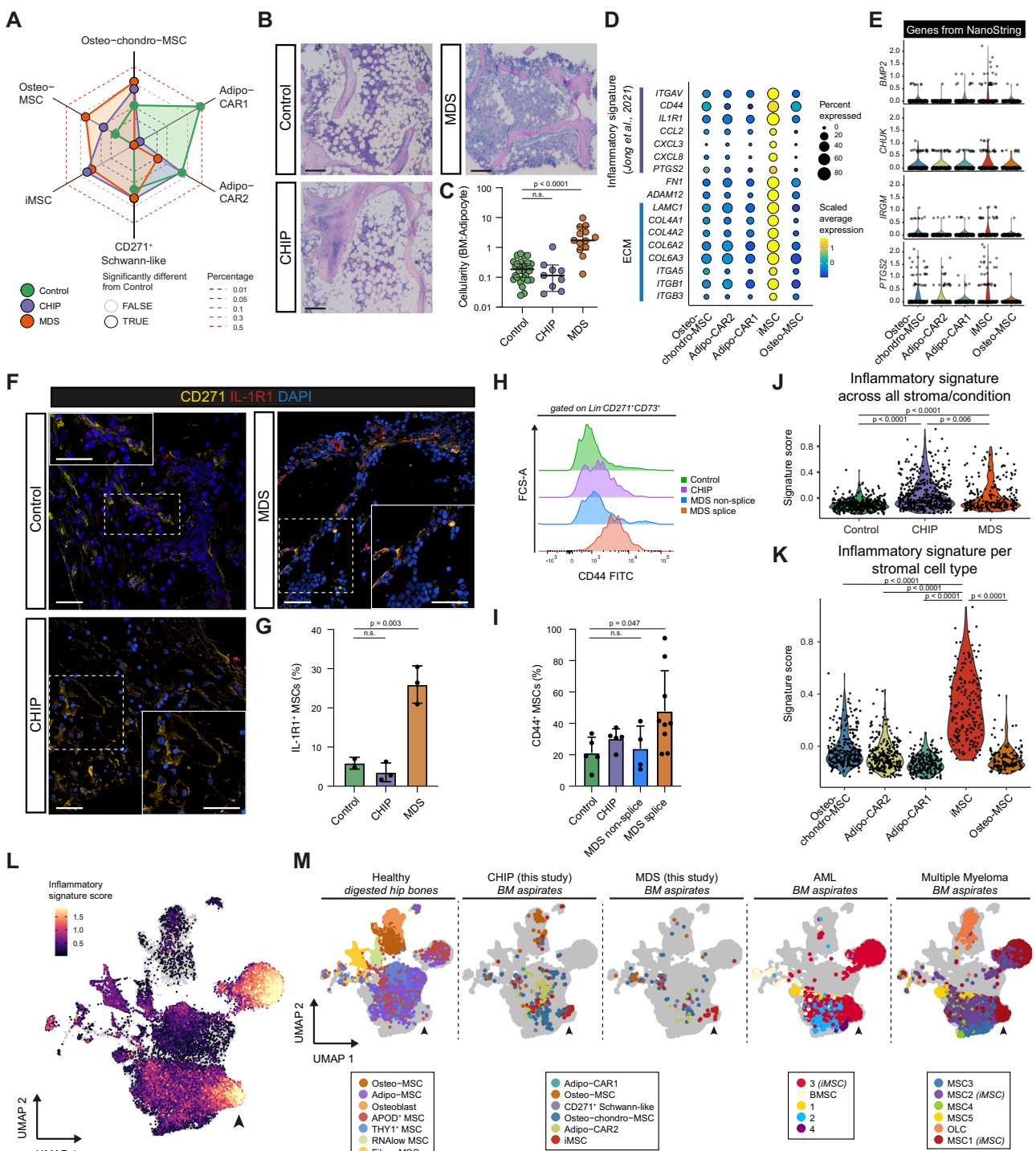

that the proportion of CXCL12-expressing CD271+ MSCs is reduced. This is consistent with the overall reduction in the HSPC-support signature in MDS stromal populations compared to Controls when pooling all MSC populations (Fig. 4D).

Notably, in CHIP-derived stromal cells, we observed a slight anti-correlation between the HSPC-support and the inflammatory signatures (Fig. 4E, PCC ≈ -0.11, *p* value ≈ 0.013, Student's *t*-test for PCC), implying that CHIP-iMSCs can provide limited HSPC support. In contrast, in the MDS stromal compartment, we observed a positive association between the inflammatory and HSPC-support signatures across all MSC populations (Fig. 4E, PCC ≈ 0.58, *p* value < 2.2e-16, Student's *t*-test for PCC), suggesting a partial retention of HSPC-supportive functions particularly the MDS-iMSCs. In line, MDS-iMSCs expressed

higher levels of *CXCL12* and *KITLG* compared to CHIP-iMSCs (Supplementary Fig. 7C), although their overall support capacity remained lower than Adipo-MSCs in Control and CHIP (Fig. 4F). A similar trend was observed in iMSCs from AML patients in the dataset from Chen and colleagues[83] (Supplementary Fig. 7D). GSEA comparing iMSCs from MDS versus CHIP revealed upregulation of stress and inflammatory pathways in MDS, including UV response, apoptosis, mTORC1 signaling, and TNFα signaling via NF-κB (Supplementary Fig. 7E), indicating MDS-iMSCs have heightened inflammatory activation in comparison to CHIPiMSCs. These observations suggest that, although MDS shows reduced HSPC-supportive capacity alongside iMSC expansion, both inflammatory and HSPC-support programs may still partially coexist within individual stromal cells.

**Fig. 3 | Identification of a distinct inflammatory MSC population in BM of CHIP and MDS. A** Spider plot showing stromal cell types in Control (green), CHIP (purple), and MDS (orange). Proportions are probit-transformed; black outlines indicate significant differences compared to Control. Statistical test: Fisher's exact test. **B** Giemsa-stained BM sections from Control, CHIP, and MDS patients. Scale bars: 100 μm. **C** Scatter plot quantifying BM cellularity (cell/adipocyte ratio) across conditions (Control $n = 30$, CHIP $n = 9$, MDS $n = 16$ patients) (see Supplementary Fig. 5A–C). Medians with 95% confidence intervals are shown. Statistical test: one-way ANOVA, Tukey's test. **D** Dot plot of extracellular matrix (ECM)-related and inflammatory gene markers (from[74]) across stromal cell populations. Circle size represents percentage of cells expressing the gene, while color intensity represents scaled expression. **E** Violin plots of stromal genes upregulated in MDS and detected in the NanoString data. **F** Immunofluorescence of BM sections showing IL-1R1 (red) and CD271 (yellow) co-staining in Control, CHIP, and MDS. Insets highlight areas containing inflammatory IL-1R1$^+$CD271$^+$ stromal cells. DAPI (blue) stains nuclei. Scale bars: 25 μm. **G** Bar graph quantifying IL-1R1$^+$ MSCs across Control ($n = 2$), CHIP ($n = 3$), and MDS ($n = 3$) samples. Medians with 95% confidence intervals are shown.

Statistical test: one-way ANOVA, Tukey's test. **H**, **I** Flow cytometry of CD44 expression on Lin$^-$CD271$^+$CD73$^+$ MSCs from Control ($n = 5$), CHIP ($n = 5$), and MDS ($n = 13$) samples. MDS samples are divided into splice ($n = 9$, *SF3B1*, *SRSF2*) and non-spliceosome ($n = 4$) mutated. Histogram (**H**) and percentage of CD44$^+$ MSCs (**I**). Means with SD are shown. Statistical test: one-way ANOVA, Dunnett test. **J**, **K** Violin plots of inflammatory signature scores in stromal cells, stratified by condition (**J**) and cell subtypes (**K**). The signature score represents the cumulative expression of inflammation-related genes. Statistical test: two-sided Wilcoxon rank-sum test. **L** UMAP integration of our scRNA-seq dataset with three published human BM datasets (healthy, AML, and MM[47,74,83]) using Harmony[177]. The stromal cells are colored by inflammatory signature scores. Arrowhead marks iMSCs. **M** UMAP of stromal cell populations in Control, CHIP, and MDS from BM aspirates (left), healthy digested hip bones (middle)[47], and AML[83] and MM BM aspirates (right)[74]. Cluster labels follow original published annotations, inflammation-associated clusters are labeled iMSC. **A**, **J**, **K** *P* values were adjusted using Benjamini−Hochberg procedure. Source data are provided in the Source Data file and Supplementary Data 7.

---

To assess potential stromal-HSPC interactions, we applied the computational tool NICHES[87] to infer ligand-receptor activity between MSCs subsets and HSPCs. In Controls, Adipo-CAR1 cells emerged as the primary HSPC-interacting cell population (Fig. 4G), consistent with their known supportive role (Fig. 4A). In CHIP, this interaction shifted mainly toward Adipo-CAR2, while in MDS the iMSC population became the dominant stromal population to interact with HSPCs (Fig. 4G). Notably, these iMSC-HSPC interactions were significantly more frequent in MDS than CHIP (OR ~ 3.3; *p* value < 2.2e-16), supporting that iMSCs in CHIP and MDS, though transcriptionally similar, are functionally distinct.

Altogether, these results suggest that while both CHIP and MDS display inflammatory stromal remodeling, their iMSCs differ in functional output: iMSCs in CHIP show limited hematopoietic support, whereas iMSCs in MDS retain partial capacity to interact with and support HSPCs.

## MDS blasts contribute to iMSC remodeling

Next, we sought to understand how clonal HSPCs may contribute to stromal remodeling and the emergence of iMSCs. First, we analyzed the HSPC composition across Control, CHIP, and MDS donors. The most notable change was the significant reduction of Pre-B/Pro-B cells in both CHIP and MDS compared to Control (Fig. 5A, Supplementary Fig. 2I), consistent with the diminished (mature) B cell signatures seen in our NanoString data and the reduced frequencies of CD10$^+$/CD19$^+$ B cells detected by flow cytometry (Fig. 1E, F, and Supplementary Fig. 1C). Furthermore, HSC/MPPs were slightly but significantly reduced in MDS while showing a modest increase in CHIP. Apart from that, no HSPC population displayed a marked expansion in CHIP or MDS, with the exception of an expansion of CD14-low monocytes, previously reported in other blood malignancies such as B cell leukemia[69].

Differential expression in all HSPCs comparing MDS versus Control followed by GSEA revealed a significant enrichment for pro-inflammatory pathways, such as TNFα via NF-κB in MDS (Supplementary Fig. 8A, B). Among the individual HSPC populations, this upregulation was most pronounced in GMPs and LMPPs, suggesting these populations may drive the overall enrichment (Supplementary Fig. 8A). While CHIP HSPCs showed a similar trend, the difference did not reach statistical significance (Supplementary Fig. 8A).

To further investigate lineage-specific contribution to inflammation, we applied Monocle3 to infer the HSPC differentiation trajectories (erythroid, myeloid, and lymphoid)[88] and quantified specific pro-inflammatory gene expression across pseudotime (Fig. 5B). We found that CHIP-derived HSPCs expressed higher levels of pro-inflammatory genes CX*CL8* and *IL1β* across all lineages compared to Controls, suggesting a broader low-grade inflammatory state. In MDS,

HSPCs showed moderate increases in *CXCL8* expression across all lineages, with a pronounced increase specifically in the monocyte population (Fig. 5C).

To determine whether clonal *SF3B1* MDS HSPCs themselves contribute to the inflammatory environment, we applied SpliceUp[89], our custom algorithm that infers mutational status based on the presence of characteristic mis-splicing events associated with *SF3B1* mutations (**Methods**; $n = 3$, Fig. 5D). By aggregating data across multiple splice sites, SpliceUp circumvents the sparsity inherent in scRNA-seq reads, providing improved sensitivity compared to direct mutant allele detection. While a minority of *SF3B1*-mutant cells may still be identified as wild-type due to residual sparsity, this approach enabled reliable classification of *SF3B1* mutant (*SF3B1$^{MUT}$*) and wild-type (*SF3B1$^{WT}$*) HSPCs, offering a robust framework for downstream comparative analyses.

We performed differential gene expression analysis between predicted *SF3B1$^{WT}$* and *SF3B1$^{MUT}$* HSPCs across the 3 *SF3B1*-mutated MDS donors in a pseudo-bulk approach. This analysis included primarily MDS-derived erythroid progenitor cells that were inadvertently enriched during FACS-based stromal isolation. (Supplementary Fig. 4, Fig. 5D, and Supplementary Fig. 8C). Overall, we identified 61 differentially expressed genes between *SF3B1$^{MUT}$* and *SF3B1$^{WT}$* cells (Fig. 5E), none of which were significantly upregulated in the MDS bulk BM transcriptomic data (Fig. 1E), suggesting that the *SF3B1$^{MUT}$* HSPCs contribute minimally to the inflammatory milieu within the MDS. Moreover, GSEA analysis revealed that *SF3B1$^{MUT}$* cells upregulated pathways involved in cytoplasmic translation, cellular respiration, and nucleoside metabolism, while *SF3B1$^{WT}$* cells enriched pathways are related to cell differentiation, signaling, and glycerolipid metabolism (Fig. 5F). This suggests that *SF3B1$^{MUT}$* cells exhibit an adaptation towards increased cell proliferation in lieu of terminal differentiation capacity, a hallmark of altered HSPC kinetics, which aligns with previous studies identifying this shift as a defining characteristic of MDS[90,91].

To functionally investigate how mutated HSPCs influence stromal cells, we established a co-culture system using primary BM-derived MSCs and CD34$^+$ HSPCs isolated from BM aspirates from donors of each group (Control, CHIP, MDS, $n = 3$/condition, all from the study cohort) (Fig. 5G, Supplementary Fig. 9A). Cells were cultured together for 96 hrs, and secreted proteins in the culture supernatants were profiled using the Olink platform, focusing on factors related to hematopoietic support and inflammation (Supplementary Data 6, Supplementary Fig. 9B).

After 96 hrs, Control and CHIP HSPCs formed numerous cobblestone areas, which are clusters of HSCs/MPPs, while MDS HSPCs produced very few (Supplementary Fig. 9A). In the secretome, the most striking observation was that CXCL12, along with other hematopoietic

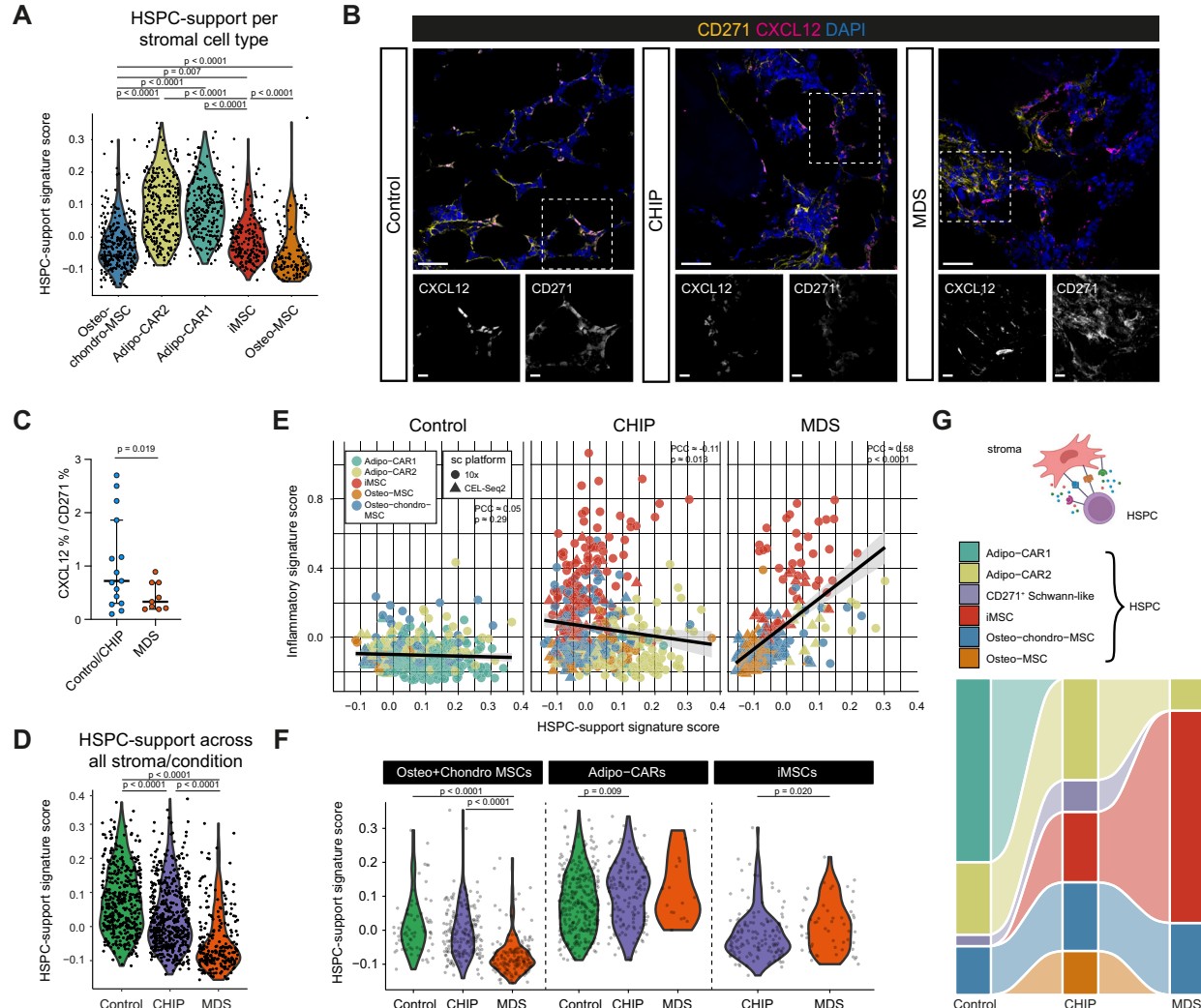

**Fig. 4 | Mapping HSPC support in stromal BM populations. A** Violin plots of HSPC-support signature scores across the stromal cell types.
**B** Immunofluorescence of BM sections showing CXCL12 (magenta) and CD271 (yellow) expression in Control, CHIP, and MDS. Insets highlight regions with distinct co-localization patterns. DAPI (blue) stains the nuclei. Scale bars: 25 µm. **C** Dot plot of CXCL12+ cells among CD271+ stromal cells (Control/CHIP $n = 15$ and MDS $n = 9$ donors). Medians with 95% confidence intervals are shown. Statistical test: two-sided Welch's *t*-test. **D** Violin plots of HSPC-support signature scores in stromal cells from scRNA-seq across Control, CHIP, and MDS. The signature score represents the cumulative expression of the HSPC-support. **E** Scatter plots showing the correlation between the HSPC-support and inflammation signature scores across stromal cells between conditions. Different colors represent the distinct stromal

clusters and different shapes represent the single-cell datasets (circles 10x, triangles CEL-Seq2). Black trend lines indicate the direction of the correlation in each condition. Linear fit and 95% confidence intervals are calculated by the function *geom_smooth* from *ggplot2*. Statistical test: student's *t*-test for PCC. **F** Violin plots of the HSPC-support signature scores specifically in Osteo-chondro MSCs, Adipo-CAR1/2, and iMSCs across conditions. **G** Sankey diagram summarizing the proportions of NICHES-inferred cell-cell interactions[87] between HSPCs and stromal cell types across Control, CHIP, and MDS. Interaction proportions highlight the changes of supportive interactions in MDS. **A**, **D**, **F** Statistical test: two-sided Wilcoxon rank-sum test with Benjamini–Hochberg procedure. Stroma-HSPC schematic created in BioRender. Prummel, K. (https://BioRender.com/8aemiyg). Source data are provided in the Source Data file.

support factors such as M-CSF (CSF1), GM-CSF (CSF2) and CCL2, were strongly induced in the co-cultures with Control and CHIP-derived HSPCs but remained at baseline levels in co-cultures with MDS-derived HSPCs (Fig. 5H). scRNA-seq of the co-cultures confirmed that these HSPC-supportive mediators were predominantly expressed by the stromal cells, except for *CCL2* (Fig. 5I). This is consistent with our earlier observation of overall reduced CXCL12 in primary MDS patient samples at both the RNA and protein levels (Fig. 4B,C).

Consistent with our observations in primary BM (Fig. 5C,E-F, Supplementary Fig. 9A), MDS HSPCs did not induce inflammatory programs in stromal cells (Supplementary Fig. 9B), and HSPC intrinsic expression of pro-inflammatory cytokines such as IL-1β, CXCL8, and various interferons were even lower in MDS compared to healthy HSPCs (Supplementary Fig. 9B).

To model a more advanced disease stage, we also co-cultured BM-MSCs with the MDS-L cell line, which represents a more differentiated, blast-like MDS population (Supplementary Fig. 10A). Secretome profiling using the Luminex platform revealed that, in contrast to MDS HSPCs, MDS blasts induced inflammatory factors such as IL-1α, MIP1, IL-1RA, while also suppressing key hematopoietic and angiogenic-support factors including CXCL12, VEGFA, and HGF (Supplementary Fig. 10B,C). These findings suggest a disease-stage-specific modulation of stromal function.

Overall, these results imply that in healthy BM, HSPCs actively engage with the stromal niche to establish supportive feedback loops, a function that appears compromised in MDS-HSPCs. In addition, MDS blasts seem to induce a pro-inflammatory environment that may favor their malignant survival and contribute to disease progression.

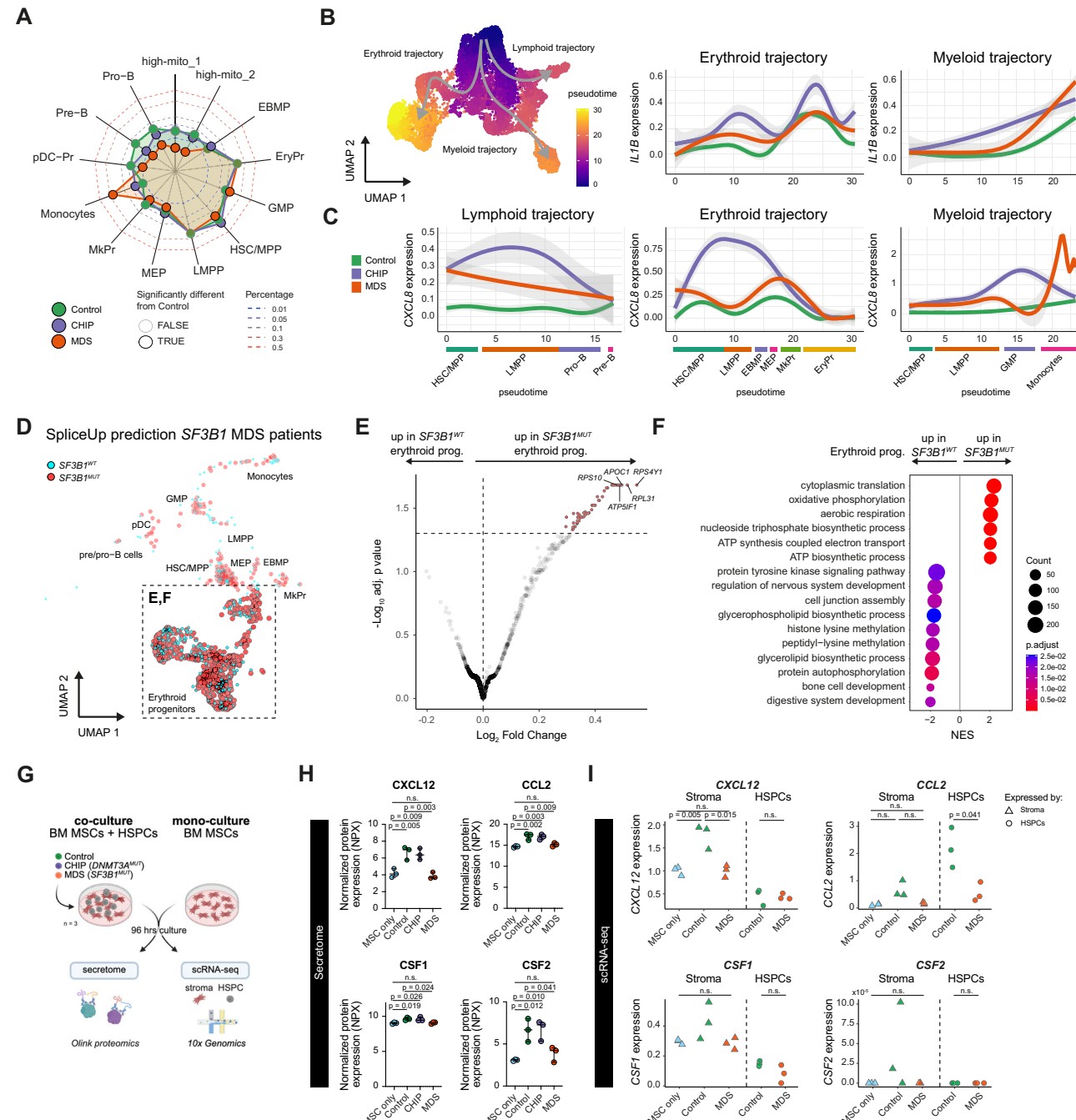

**Fig. 5 | Characterization of HSPCs and their effect on BM inflammation. A** Spider plot of the relative abundances of HSPC subtypes across Control (green), CHIP (purple), and MDS (orange) donors. Proportions are probit-transformed; black outlines indicate significant differences compared to Controls. Statistical test: Fisher's exact test. **B** UMAP of HSPC cells showing the pseudotime and schematic outline of main differentiation trajectories. **C** Smoothed expression of pro-inflammatory cytokine genes *CXCL8* and *IL1B* along the pseudotime in 3 main differentiation trajectories of HSPCs. Smoothing and 95% confidence intervals are calculated by the function *geom_smooth* from *ggplot2* (posterior distribution of coefficients of a cubic spline generalized additive regression model used for smoothing). **D** UMAP of HSPCs showing the SpliceUp-predicted *SF3B1* mutational status (*SF3B1^WT^* blue, *SF3B1^MUT^* red). Dashed box highlights the erythroid progenitors, further analyzed in (**E**, **F**). **E** Differential expression between SpliceUp-predicted *SF3B1^WT^* and *SF3B1^MUT^* erythroid progenitors. Statistical test: DESeq2. Red dots: adjusted *p* value < 0.05. **F** Gene Set Enrichment Analysis (GSEA) results with the Biological Process ontology from Gene Ontology database showing the top 20 significant sets enriched between *SF3B1^WT^* and *SF3B1^MUT^* erythroid progenitors.

**G** Schematic visualization of the primary HSPC-MSC co-culture experiment. BM MSCs were cultured either alone or with CD34⁺ HSPC from Control, CHIP or MDS donors (ratio 1:1). After 96 hrs, the supernatant was collected and subjected to subsequent protein quantification using an Olink panel. The cells were harvested and subjected to scRNA-seq. Created in BioRender. Prummel, K. (https://BioRender.com/v0w0jhd). **H** Olink quantification of 4 secreted cytokines after 96 hrs of co-culture. Mean normalized protein expression (NPX) values (mean of 3 experimental replicates) across conditions (MSC mono-culture (MSC-only), Control CHIP, and MDS, *n* = 3/group) are displayed. Medians with 95% confidence intervals are shown. Statistical test: one-way ANOVA, Tukey's test. **I** scRNA-seq of co-cultures showing normalized RNA expression of cytokines in (**H**) across the stromal (triangles) and HSPC (circles) compartments after MSC mono-culture (MSC-only), and Control and MDS co-cultures (*n* = 3/group). Medians with 95% confidence intervals are shown. Statistical test: two-sided unpaired Student's *t*-test. **A**, **E**, **F** *P* values were adjusted using Benjamini–Hochberg procedure. Source data are provided in the Source Data file and Supplementary Data 7. n.s. not significant.

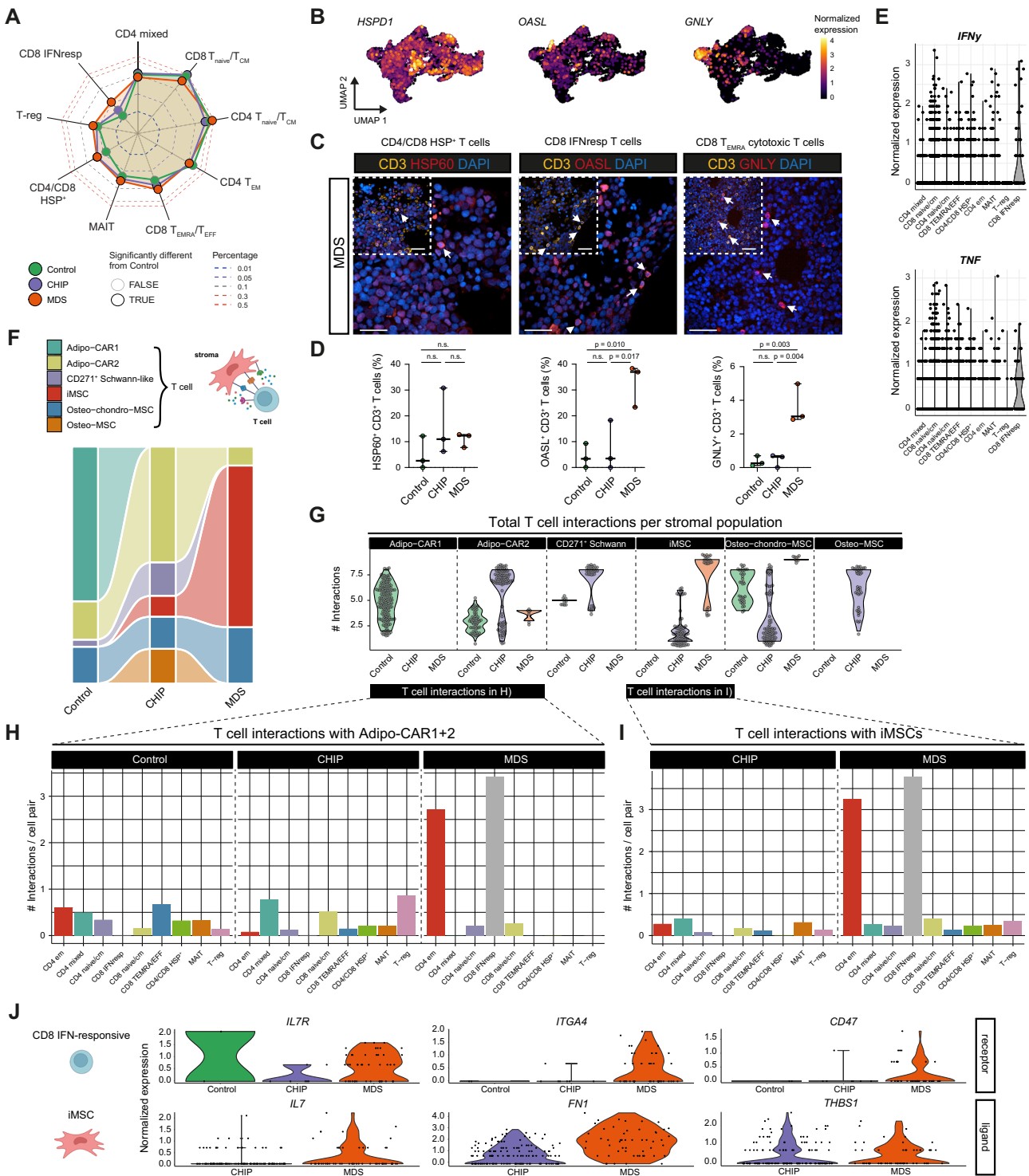

**MDS-specific IFN-responsive T cells interact with iMSC in MDS**

To better understand the immune contributors to the inflammatory signature in CHIP and MDS BM, we investigated the T cell landscape in our dataset. Three distinct T cell populations were significantly differentially abundant in Controls: Heat Shock Protein (HSP)$^+$ T cells were elevated in CHIP and MDS, while IFN-responsive and cytotoxic CD8$^+$ T$_{EMRA}$/T$_{EFF}$ cells were specifically enriched in MDS (Fig. 6A, Supplementary Fig. 4J). In addition, a modest but significant increase in Tregs was observed in MDS (Fig. 6A), consistent with our immunophenotyping data based on CD4 and FoxP3 expression (Supplementary Fig. 1D). While HSP$^+$ T cells have been previously described in

colorectal cancer, their biological function remains unclear[92]. In contrast, both IFN-responsive and cytotoxic CD8$^+$ cells were recently associated with exacerbated inflammation in autoimmune diseases such rheumatoid arthritis[93–95].

We validated the expansion of these inflammation-related T cell subsets in BM using immunofluorescence of CD3$^+$ T cell compartment with previously reported markers also expressed in our scRNA-seq data on a subset of our cohort. We used OASL for IFN-responsive T cells[96], GNLY for cytotoxic T$_{EMRA}$/T$_{EFF}$ cells[97], and HSP60 for HSP$^+$ T cells[98] (Fig. 6C, and Supplementary Fig. 11). The OASL-expressing IFN-responsive and GNLY-expressing cytotoxic T$_{EMRA}$/T$_{EFF}$ cells showed a

**Fig. 6 | Characterization of T cells in BM of Control, CHIP, and MDS. A** Spider plot of relative T cell population abundance across Control (green), CHIP (purple), and MDS (orange) donors. Proportions are probit-transformed; black outlines indicate significant differences compared to Controls. Statistical test: Fisher's exact test with Benjamini–Hochberg procedure. **B** UMAPs showing the normalized expression of genes enriched in CHIP and/or MDS-associated T cells subsets: CD4/CD8 HSP+, CD8+ T_EMRA, and CD8+ IFN-responsive. **C** Immunofluorescence of MDS BM sections showing HSP60, OASL, and GNLY (red; left to right) representing the CD4/CD8 HSP+, CD8 IFN-responsive T cells, and CD8+ T_EMRA, respectively, with CD3 (yellow). Insets highlight co-localization. DAPI (blue) stains the nuclei. Arrows indicate double-positive cells. Scale bars: 25 μm. **D** Quantification of marker-positive cells from (**B**) among all CD3+ cells (n = 3/condition). Medians with 95% confidence intervals are shown. Statistical test: one-way ANOVA, Tukey's test.

**E** Violin plots of *IFNy* and *TNF* gene expression across T cell subsets. Each dot represents the expression level of an individual cell. **F** Sankey diagram summarizing the proportions of NICHES-inferred cell-cell interactions between T cells and different stromal cell types across Control, CHIP, and MDS samples. Stroma-T cell schematic created in BioRender. Prummel, K. (https://BioRender.com/wg4r69i). **G** Numbers of interactions between T cells (receiver) per individual stromal cell (sender) inferred by NICHES. Point jitter is added to visualize overlapping points. **H**, **I** Total interactions between Adipo-CARs (**H**) or iMSCs (**I**) and T cell subtypes (receivers) across conditions. Numbers of interactions are normalized by the number of all possible interactions given the population sizes. **J** Expression of ligand-receptor pairs marking the interactions between iMSCs and CD8+ IFN-responsive cells. Source data are provided in the Source Data file and Supplementary Data 7. n.s. not significant.

---

significant increase in MDS (Fig. 6D), confirming our scRNA-seq observations. In CHIP, the expansion of HSP+ T cells was visible but not statistically significant (Fig. 6D). These data confirmed that the T cells analyzed in the scRNA-seq data were not solely circulating T cells present in the BM vasculature.

IFN-responsive T cells showed elevated transcript levels of pro-inflammatory cytokines TNFα and IFNγ in the normalized scRNA-seq data (Fig. 6E), suggesting they may contribute to immune activation through self- or cross-activation mechanisms, thereby functioning as potent drivers of inflammation. Additionally, we previously identified a similar IFN-responsive T cell population in the BM of AML patients after allogeneic stem cell transplantation associated with imminent relapse[65], suggesting these T cells may be ineffective in mounting anti-leukemic responses. Furthermore, comparison of the bulk BM expression signature upregulated in MDS versus Control in our NanoString data revealed that IFN-responsive T cells highly expressed genes involved in the inflammatory response (*IFIT1/2/3*, *TNF*; Fig. 2I). Together, these findings suggest that aberrant IFN-responsive CD8 T cells might be key contributors to the inflammatory milieu in MDS.

Using NICHES[87], we investigated potential interactions between stromal populations and T cells. In both Control and CHIP samples, the Adipo-CAR1 and Adipo-CAR2 populations exhibited the most interactions with T cells respectively, whereas in MDS, iMSCs were the major interacting stromal population (Fig. 6F, G). Furthermore, the T cell interactions in MDS differed quite dramatically from Control and CHIP where Adipo-CARs interacted with all CD4/CD8 populations: mostly with Tregs, CD4_naive/CM, and CD4 mixed cells in CHIP, while in MDS, their interactions were primarily with IFN-responsive T cells and CD4_EM cells (Fig. 6H). Finally, while MDS-iMSCs strongly interacted with IFN-responsive and CD4_EM T cells, CHIP-iMSCs showed very little interactions with any T cell population (Fig. 6I), corroborating the observation that the iMSCs in CHIP and MDS represent two distinct functional populations.

We investigated the specific ligand-receptor interactions mediating the crosstalk between iMSCs and T cells in MDS. In particular, we identified several pairs enriched in MDS, including *IL7*, expressed by iMSCs, interacting with *IL7R* (*CD127*) on IFN-responsive T cells. Both the ligand (expressed by iMSCs) and receptor (on T cells) are upregulated in MDS compared to CHIP (Fig. 6J). IL-7R expression can be induced by IFN signaling and IL-7 is a regulator of T cell differentiation and homeostasis[99]. Other prominent ligand-receptor interactions included Fibronectin 1 (*FN1*), secreted by iMSCs, interacting with Integrin 4A (*ITG4A*) on T cells, as well as Thrombospondin 1 (*THBS1*), an extracellular matrix glycoprotein produced by iMSC, interacting with *CD47* on T cells (Fig. 6J).

These results indicate that IFN-responsive T cells may contribute to the inflammatory signaling via TNFα and IFNγ secretion, while the blast-induced ECM remodeling of iMSCs in MDS may specifically attract inflammatory T cell populations.

## Aberrant cross-talk of MSCs and mutated HSPCs promotes vascular niche remodeling in MDS

Our integration of the targeted bulk RNA-seq data with healthy BM indicated elevated expression of inflammation-associated genes in endothelial cells in MDS (Fig. 1G). Previous studies have described expansion and remodeling of BM vasculature and increased expression of angiogenic factors including *VEGFA* in myeloid malignancies, including MDS[100–103]. Due to the limited availability of endothelial cells in human BM aspirates and the challenges in capturing them through FACS and scRNA-seq, we assessed the angiogenic potential of BM cell types: HSPCs, T cells, and stromal populations (Fig. 7A, and Supplementary Fig. 12A–C). We found that the Adipo-CAR and iMSC populations were primarily responsible for angiogenesis-related condition-dependent changes (Fig. 7A, Supplementary Fig. 12C), while other cell populations showed minimal angiogenic potential (Supplementary Fig. 12A, B). This is in line with the reported localization of Adipo-CARs in vascular BM niches in mouse and human BM[47,104–106]. Specifically, Adipo-CARs in MDS patients upregulated several known secreted vasculature-remodeling factors, including, *VEGFA* and *FSTL1*, which were also secreted by iMSCs both in CHIP and MDS (Fig. 7B). These results suggest that on top of increased inflammation and reduced HSPC-support, the remodeled stroma in MDS may reshape the BM vasculature. We leveraged immunofluorescence imaging on FFPE BM biopsies to investigate this increased angiogenic potential within the MDS niche on the arteriolar and sinusoidal blood vessels (Fig. 7C). We quantified the microvasculature density (MVD) for both arteriolar (UEA1+CD105-) and sinusoidal (UEA1-CD105+) vasculature, which revealed a significant increase in MVD for both structures in MDS compared to Control and CHIP (Fig. 7D, E). Additionally, we observed a reduction in the average distance between sinusoidal and arteriolar vessels in MDS compared to CHIP and Control (Fig. 7F).

Despite the presence of iMSCs with angiogenic potential within CHIP donors (Fig. 7A, B), no significant vascular changes were observed (Fig. 7C–F). Given the correlation between osteoporosis risk and a higher VAF% of the CHIP mutation *DNMT3A*^R882H (>10%)[107,108], we further explored the yet uncharted impact of VAF on vasculature remodeling. We utilized a CHIP mouse model carrying the *Dnmt3a*^R878H mutation, which is homologous to a common human *DNMT3A*^R882H CHIP mutation, where the majority of hematopoietic cells carry this mutation[109]. We analyzed the general BM morphology and vasculature within femurs of young control mice (24 weeks), young *Dnmt3a*^R878H/+ mice (24 weeks), and old control mice (68 weeks) (Fig. 7G). First, we showed that *Dnmt3a*^R878H/+ decreased the overall cellularity in BM, driven by the expansion of adipocytes particularly in the femoral head (Fig. 7H,I). This phenotype may reflect impaired bone integrity, consistent with prior reports of increased marrow adiposity and bone loss in *Dnmt3a*-mutant CHIP mouse models[107,110]. Next, we observed that young *Dnmt3a*-mutant mice showed increased sinusoidal vessel density (Fig. 7J, L) and general morphological alterations including increased dilation of the sinusoidal vasculature (Fig. 7K), which is in

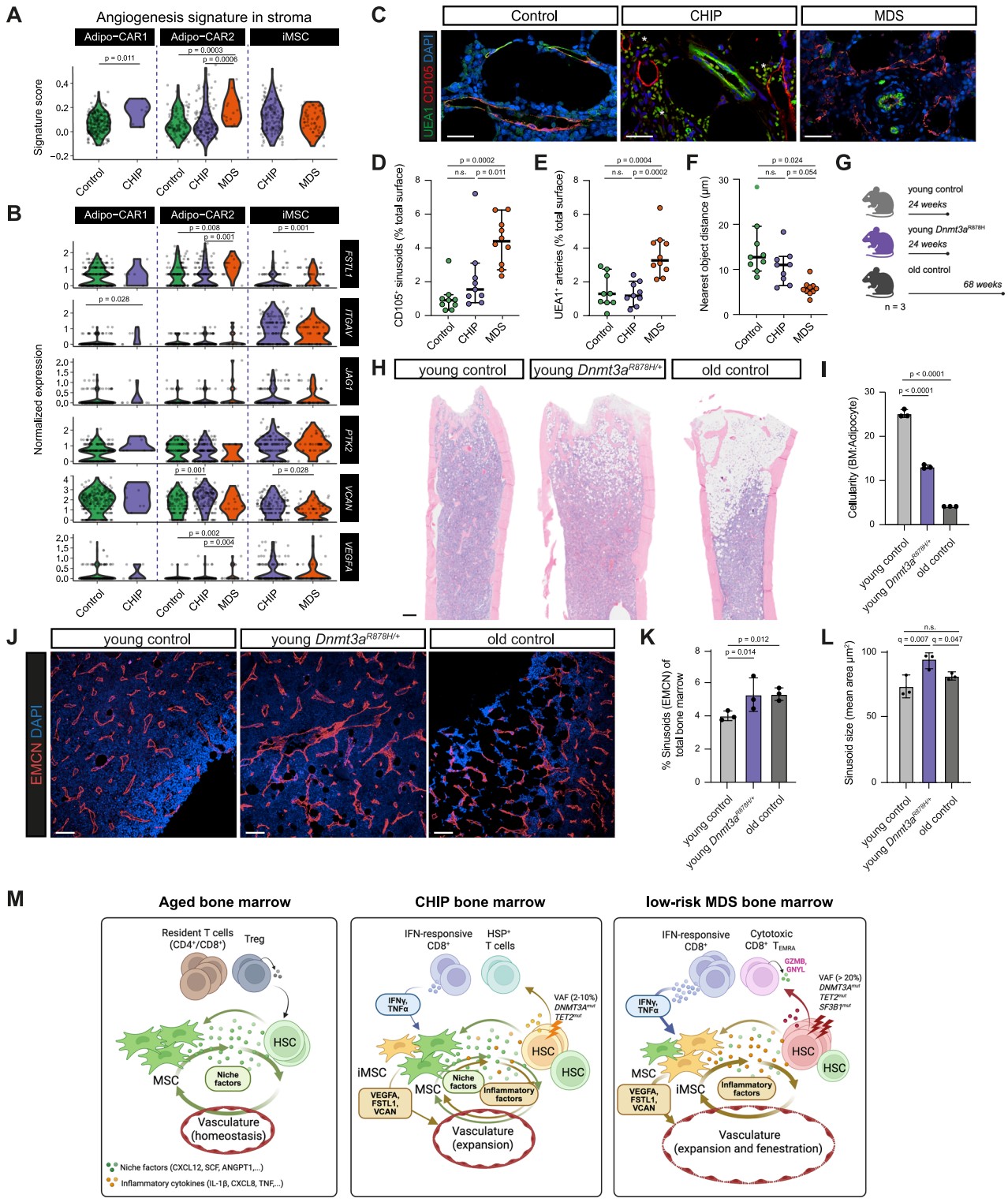

line with our observations in human MDS (Fig. 7C-F). Arterioles and arteries did not show any significant differences between conditions (Supplementary Fig. 12D). This suggests that the higher burden of mutant cells in CHIP may drive BM niche architecture remodeling, however, additional studies in human individuals with a higher CHIP VAF will be needed to validate this.

Altogether, these results suggest that mutant HSPCs can significantly alter the BM niche vasculature, potentially via stromal remodeling. These findings provide new insights into how CHIP may prime the BM microenvironment for malignant transformation.

## Discussion

Prior culture-based and bulk transcriptomic studies have suggested an inflammatory skewing of MSCs in aging and chronic BM disorders[42,111–115]. Our study builds upon these observations by providing a comprehensive single-cell characterization of the BM microenvironment across healthy aged controls, CHIP, and MDS, revealing inflammatory remodeling of the hematopoietic BM niche, characterized by coordinated stromal and immune perturbations (Fig. 7M).

A central finding is the emergence of a distinct inflammatory mesenchymal stromal cell (iMSC) population, absent in healthy

**Fig. 7 | Assessment of vascular alterations in human CHIP, MDS, and a CHIP *Dnmt3a* mouse model. A** Violin plots of angiogenesis signature scores across stromal cell types comparing Control, CHIP, and MDS. **B** Violin plots of representative genes within the angiogenesis signature across the stromal cell types comparing Control, CHIP, and MDS. **C** Immunofluorescence of BM sections showing UEA1 (green, all vasculature) and CD105 (red, sinusoids) and DAPI (blue) across Control, CHIP, and MDS. Asterisks are highlighting autofluorescence from erythrocytes. Scale bars: 25 μm. **D, E** Quantification of CD105$^+$ sinusoids (**D**) and CD105$^-$UEA1$^+$ arterioles/arteries (**E**) across conditions ($n = 3$/condition). Medians with 95% confidence intervals are shown. Statistical test: one-way ANOVA, Tukey's test. **F** Scatter plot depicting the nearest distance of the vasculature (in μm) in Control, CHIP, and MDS samples. Shortened distances in MDS indicate disrupted vasculature organization. Medians with 95% confidence intervals are shown. Statistical test: one-way ANOVA with FDR correction. **G** Schematic of the CHIP *Dnmt3a*-mutant mouse study groups: femurs were collected from young control (24 weeks), young *Dnmt3a*$^{R878H}$ (24 weeks), and old control (68 weeks), $n = 3$/group. Created in BioRender. Prummel, K. (https://BioRender.com/pqyu7b8). **H** Giemsa staining from young control, young *Dnmt3a*$^{R878H}$, and old control mouse femurs, illustrating reduced cellularity in young *Dnmt3a*$^{R878H}$ and old mice ($n = 3$/group). Scale bars: 200 μm. **I** Bar plot showing the cellularity (BM/adipocyte ratio) across the groups ($n = 3$/group). Both young *Dnmt3a*$^{R878H}$ and old control mice display significantly reduced cellularity compared to young controls. Means with SD are shown. Statistical test: one-way ANOVA, Tukey's test. **J** Immunofluorescence of EMCN (red) and DAPI (blue) in femur sections from young control, young *Dnmt3a*$^{R878H}$, and old control mice. Scale bars: 100 μm. **K** Quantification of the total sinusoid area in the femur ($n = 3$/condition). Means with SD are shown. Statistical test: two-way ANOVA, Tukey's test. **L** Quantification of the mean sinusoidal area (sinusoid size) in the femur ($n = 3$/condition). Means with SD are shown. Statistical test: one-way ANOVA with FDR correction. **M** Schematic illustration summarizing BM niche remodeling in CHIP and LR-MDS. **A, B** Statistical test: two-sided Wilcoxon rank-sum test with Benjamini–Hochberg procedure. Created in BioRender. Guezguez, B. (https://BioRender.com/8tl5dtf). Source data are provided in the Source Data file and Supplementary Data 7. n.s. not significant.

controls but expanded in CHIP and MDS. These iMSCs display a pro-inflammatory transcriptional profile and may serve as active responders to hematopoietic stress. Their presence in individuals with relatively low-risk mutations (*DNMT3A*, *TET2*, and *SF3B1*) suggests that early clonal events can coincide with - or potentially initiate - niche remodeling. Although these mutations differ in their leukemic potential[116,117], they have all been linked to inflammatory signaling: *DNMT3A*-mutant HSPCs are associated with increased levels of inflammatory cytokines and can modulate stromal cell fate[35,107,118], while *SF3B1* mutations are associated with dysregulated innate immune activation[119–122]. Emerging evidence, including work by Jakobsen et al.[35] and Scherer et al.[123], points to a more complex relationship. Rather than mutations alone driving inflammation, a pre-existing inflammatory microenvironment may act as a selective force promoting the expansion of mutant HSPCs. In line with these studies, our data supports a strong association of CHIP and MDS with an inflamed BM niche. While the origin of inflammation in non-malignant clonal states like CHIP remains uncertain, our findings suggest that MDS blasts can directly induce inflammatory remodeling of stromal cells. Moreover, a parallel study by Chen et al.[124] independently identifies iMSCs in a genetically diverse LR-MDS cohort, reinforcing the robustness and broader relevance of this inflammatory stromal phenotype.

Phenotypically, iMSCs in our study were characterized by elevated CD44 and IL-1R1 expression, both canonical NF-κB target genes induced by TNF and IL-1 signaling. These features mirror cancer-associated fibroblasts (CAFs) in solid tumors[125], where IL-1R1 has been linked to enhanced motility, inflammatory signaling, and immune suppression[126]. The shared pro-inflammatory program between iMSCs and CAFs suggests a conserved stromal response to chronic inflammatory cues. This is further supported by the skewed MSC lineage composition observed in CHIP and MDS, marked by expansion of osteo-lineage subsets and contraction of adipogenic progenitors, consistent with the lineage shift commonly associated with inflammatory microenvironments[127,128].

While both CHIP and MDS display the presence of iMSCs, our data suggest transcriptional and functional divergence between these states. In CHIP, inflammatory and hematopoietic support features within MSCs tend to be mutually exclusive (i.e., iMSCs have low support signatures), whereas in MDS these features become increasingly decoupled and can co-exist within the same iMSC population. Functional in vitro data support these observations: whereas healthy and CHIP HSPCs induced expression of stem cell-supportive factors such as CXCL12 in MSCs, MDS HSPCs and MDS-L (LR-MDS blast-like) cells failed to do so. This aligns with our single-cell data showing reduced *CXCL12* expression in MDS stroma. Notably, MDS-L also triggered inflammatory activation in MSCs, suggesting that early niche

remodeling is already occurring in LR-MDS disease. The loss of stem cell support observed in LR-MDS stroma contrasts with findings on HR-MDS by Jann et al.[129], who identified a CXCL12–high expressing MSC subset (StemCAR) enriched in response to high-risk MDS blasts and diminished following hypomethylating agent therapy. These discrepancies may reflect disease stage–specific effects on MSC function. Future studies are necessary to delineate how MSC programs diverge across the spectrum of low- and high-risk MDS.

These stromal changes are accompanied by vascular remodeling. In MDS, we observed increased angiogenic gene expression (e.g., *VEGFA*) in Adipo-CARs and iMSCs, correlating with sinusoidal expansion and increased BM permeability. This is consistent with previous reports showing that cultured MDS-derived MSCs express elevated levels of VEGFA and ANGPTL4[42]. While human low-VAF CHIP donors lacked overt vascular changes, high-VAF *Dnmt3a* CHIP mice exhibited clear vascular remodeling, including disorganized sinusoids and barrier dysfunction. This suggests that vascular remodeling in CHIP may be contingent on clonal burden, and becomes more pronounced at later stages of clonal expansion. Increased vascular permeability may facilitate aberrant HSPC activation, mobilization, and clonal dominance; thereby offering a potential mechanistic link between inflammation, vascular dysfunction, and disease progression. Spatially resolved transcriptomics and high-resolution vascular imaging will be essential to delineate how angiogenic remodeling and niche support vary across distinct anatomical compartments of the human BM in CHIP and MDS.

The iMSC expansion also coincided with selective changes in the BM T cell landscape. CHIP donors were marked by enrichment of HSP-expressing CD4$^+$/CD8$^+$ T cells, a subset not well-defined functionally but implicated in chronic inflammatory responses[130,131]. In MDS, IFN-responsive CD8$^+$ T cells were expanded and showed predicted interactions with iMSCs via IL-7/IL-7R and ITGA4/FN1 axes. This crosstalk may reinforce local inflammation and T cell retention in the BM. Additionally, the cytotoxic CD8$^+$ T$_{EMRA}$ cells present in MDS may harbor tumor-specific TCRs, which is consistent with the notion of a systemic immune response[132,133]. Thus, there is a need for increasing research efforts, including TCR-sequencing, to deepen our understanding of T cell biology and implication in CHIP and MDS.

Transcriptional profiling also revealed other dysregulated immune cell types in MDS, including non-classical monocytes, characterized by elevated expression of inflammatory markers such as *ADORA2A* and the alarmin *S100A8*. A recent preclinical study demonstrated that S100A9, secreted by monocytes/macrophages and forming a heterodimer with S100A8, can drive inflammatory activation of stromal cells and impair hematopoietic function in MDS[134]. These findings raise the possibility that monocyte-driven inflammation may contribute to the disrupted immune microenvironment in MDS,

however, further investigation is needed to clarify their functional role and interplay with stromal alterations.

The transcriptional signature of iMSCs appears conserved across hematologic malignancies. We observed strong overlap between iMSCs identified in CHIP/MDS and those reported in AML[83] and multiple myeloma (MM)[74]. Similar stromal phenotypes and vascular remodeling have also been reported in MGUS, the precursor state to MM and the lymphoid analogue of CHIP[135,136]. Together, these findings support a unifying model in which early oncogenic or clonal events - regardless of lineage - either trigger a conserved stromal inflammatory response or are selected by an inflammatory environment.

The presence of iMSCs in both premalignant (CHIP, MGUS) and malignant (MDS, AML, MM) states raises important questions about their ontogeny, plasticity, and therapeutic relevance. Whether iMSCs arise from pre-existing CAR-like populations or represent a distinct lineage under inflammatory stress remains to be resolved. Their stability and reversibility in vivo, particularly in response to anti-inflammatory or niche-targeted interventions, may inform their utility as biomarkers or therapeutic targets. Future efforts should focus on spatial mapping, clonal tracing, and stromal manipulations models to define roles of iMSCs and aberrant immune cells in early hematologic disease.

In conclusion, our study identifies inflammatory stromal remodeling as a key hallmark in both CHIP and MDS, with iMSCs emerging as key mediators. While CHIP HSPCs preserve the capacity to induce niche support, MDS HSPCs fail to do so, linking stromal dysfunction to impaired hematopoiesis in MDS. Overall, these findings underscore the role of the microenvironment in early myeloid disease and may contribute to the development of future strategies aimed at intercepting pre-malignant hematopoiesis.

## Methods
### Patient cohort and tissue material processing
Overall, BM samples from 32 MDS patients, 17 CHIP donors, and 35 healthy (Control) donors were analyzed within this study. Human BM aspirate and hip and femur trephine core biopsies were obtained from informed consenting donors enrolled in the BoHemE study (NCT02867085). All human material have been approved by local ethics committees at the University Hospital Dresden (TU Dresden, reference nr.: EK 393092016) and University Hospital Leipzig (University of Leipzig, Faculty of Medicine, reference nr.: 137/19-1k) and conducted in accordance with the Declaration of Helsinki. A detailed overview of the patient cohort, including age and sex, can be found in Supplementary Data 1. We aimed for a sex-balanced cohort and included sex-specific analyses for the NanoString data (Fig. 1) and BM cellularity (Supplementary Fig. 5B, C). Aspirates were processed by density gradient centrifugation using Ficoll-Paque Premium 1.073 (Cytiva) to deplete erythrocytes and were stored in freezing medium (90% FCS + 10% DMSO (Sigma-Aldrich)) in liquid nitrogen. Trephine cores were fixed in 4% paraformaldehyde (PFA) for 24 hrs, then transferred to PBS with sodium azide (0.3%) until further processing. Samples were decalcified for 48 hrs using Osteosoft (Merck KGaA) at 37 °C and embedded in paraffin.

### Animal models
Femurs from control and DNMT3A-R878H mutant mice were obtained from The Jackson Laboratory, USA. Briefly, *Dnmt3a*[fl-R878H/+] mice (JAX:032289) were crossed with B6.Cg-Tg(Mx1-cre)1Cgn/J mice (JAX:003556, referred to as Mx-Cre). In all experiments, control (+/+) mice carried a single copy of the Mx-Cre allele. Mice were maintained under controlled environmental conditions with a 12 hr light/12 hr dark cycle. Ambient temperature was kept at 18–21 °C, with relative humidity maintained at ~55%. To induce Mx-Cre, mice were injected intraperitoneally with 15 mg/kg high molecular weight polyinosinic-polycytidylic acid (polyI:C) (InvivoGen) once every other day for a total

of five injections. In all experiments, mice were used > 4 weeks after polyI:C administration. Following the Jackson Laboratory recommendations for aging stages, mice up to 26 weeks (6–7 months) were considered young, while mice 68 weeks (15–16 months) and older were considered old. Due to the development of ulcerative dermatitis, old *Dnmt3a*[fl-R878H/+] mice could not be included in the study. All experimental mice were female and were euthanized by cervical dislocation at 26 and 68 weeks of age for phenotypic analysis and tissue collection. Femurs were fixed in 4% PFA at 4 °C overnight and stored in PBS with sodium azide (0.3%). After decalcification, bones were paraffin embedded. The Jackson Laboratory Institutional Animal Care and Use Committee approved all experiments.

### Flow cytometry of BM aspirates
**Diagnostic flow cytometry.** Diagnostic flow cytometry was performed within 24 hrs after BM aspiration. Prior to antibody staining, erythrocytes were removed by lysis for 10 min at room temperature (RT) using BD Pharm-Lyse (1:10 dilution with distilled water; BD Biosciences), followed by two washing steps with PBS. For surface labeling, the cells were incubated with one of the five 8-color antibody panels (see Supplementary Data 3 for antibody panels) for 15 min at RT in the dark[137]. The five panels allowed a comprehensive analysis of the BM aspirates as proposed in the guidelines of the iMDSFlow[138]. Subsequently, cells were washed twice and resuspended in PBS. Samples were stored at 4 °C and measured within 1 hr. Samples were acquired on a Canto II (BD Biosciences). At least 200,000 events were acquired per sample. For data analysis, a hierarchical gating strategy according to the iMDSFlow[138] was applied: (1) exclusion of doublets (FSC-A vs. FSC-H) and (2) of debris (FSC vs. SSC), (3) gating of CD45[dim/+] leukocytes (SSC vs. CD45).

**T cells and inflammatory MSC flow cytometry.** BM aspirates from the curated BoHemE study (NCT02867085) were thawed in 100% FCS (Thermo Scientific) + 100 µg/ml DNAse I (Sigma-Aldrich), centrifuged at 300 g for 5 min and suspended in DMEM (Gibco) + 10% FCS + 100 µg/ml DNAse I. When necessary, cell pellets were treated with 1X Red Blood Cell Lysis (Invitrogen) for 10 min at RT. Cells were washed with 2% FCS/PBS. Live/dead staining (e.g., Zombie Aqua, Biolegend) was performed in PBS, followed by antibody staining for 30 min at 4 °C (see Supplementary Data 3 for antibody panels and dilutions). Stained cells were analyzed with a FACSAria Fusion (BD Biosciences) or Symphony A3 (BD Biosciences).

### Immunohistochemistry and histopathological stainings of FFPE bone marrow sections
Serial sections of human and mouse FFPE BM biopsies were prepared at 3–5 µm thickness on coated microscope slides (Dako FLEX, Agilent) and processed for immunohistochemistry/immunofluorescence (IHC-IF). Deparaffinization (30 min), rehydration (2×10 min), and antigen retrieval (10 min) were performed by heating sections in Trilogy™ buffer (Cell Marque, Millipore Sigma) at 105 °C and 1.2 bar for 10 min in a pressure cooker. Sections were permeabilized with 0.25% Triton-X (Sigma) in PBS for 10 min, then blocked for 30 min with 5% normal donkey serum (Jackson ImmunoResearch) in PBS-Tween20 (0.05%, Sigma). Primary antibodies were incubated at RT for 1 hr in 1% normal donkey serum PBS-Triton-X100 (0.2%). Samples were washed twice with 1% donkey serum PBS-Tween20 (0.05%) for 5 min, then incubated with secondary antibody for 1 hr at RT. Samples were washed again with 1% donkey serum PBS-Tween20 (0.05%) for 5 min, then incubated for 3 min with the TrueView (Vector Laboratories) quenching reagent to reduce tissue autofluorescence. Slides were washed with PBS for 5 min, mounted with DAPI-containing mounting medium (Abcam), and imaged. All used antibodies in IHC-IF are detailed in Supplementary Data 3. Additional FFPE sections of BM biopsies were processed for

standard pathology diagnosis with Giemsa staining by the Core Facility Biobank of the UMC Mainz.

## Image data acquisition and analysis

IHC-IF stained sections were imaged for semi-quantitative analysis on an Opera Phenix (PerkinElmer) high-content screening system with 40x objective (water, NA 1.1). Image analysis was done using Harmony 4.8 (PerkinElmer), detailed workflows for image analysis can be found in Supplementary Fig. 5.

Giemsa-stained sections were imaged using an EVOS M5000 microscope with 4x objective (air, NA 0.13). Captured images of human BM biopsy sections were analyzed for cellularity analysis using the Weka-Segmentation plugin for Fiji; for each condition, one representative image was used as training data[139]. For each sample, three different fields of views were analyzed and the resulting probability maps were checked for successful segmentation. The detailed workflow can be found in Supplementary Fig. 5. For the Giemsa staining of mouse femurs, cellularity analysis was conducted using QuPath software with the MarrowQuant plugin[140].

## Co-culture experiments

**HSPC – MSC co-cultures.** Primary BM-derived MSCs were isolated from a healthy young male donor (32 years old) and used at passage 1 (P1), after informed consent and in accordance with procedures approved by the local ethics committee (Comité de Protection des Personnes–Île-de-France V, Hôpital Saint-Antoine, Paris) and with the Declaration of Helsinki. For secretome and scRNA-seq analysis, MSCs were seeded at a density of 30,000 cells per well in gelatin-coated 12-well plates (Costar, Corning Incorporated) with 1 ml Myelocult H5100 medium (Stemcell Technologies) with 1 μM Hydroxycortisone (Sigma). After 24 hrs, HSPCs were isolated from BM aspirates using CD34+ magnetic bead–based positive selection (Miltenyi Biotec) and added to the MSCs at a 1:1 ratio (30,000 HSPCs per well). Co-cultures were maintained for 96 hrs under standard conditions (37 °C, 5% $CO_2$, > 90% humidity). For each donor, three technical replicates were set up. Additional control wells containing MSC mono-cultures and medium-only wells (Myelocult in gelatin-coated plates without cells) were included in each experiment. Three independent rounds of co-culture experiments were performed, each including one donor per condition (Control, CHIP, and MDS) to minimize batch effects, resulting in three biological replicates per condition (see Supplementary Fig. 9A). After 96 hrs, brightfield images were acquired with a Nikon Eclipse TS100 (with 4x (X4, X0.13, WD 16.5) and 40x (X40, X0.55, WD 2.1) objectives) to capture the culture morphology.

Next, cells were trypsinized (Trypsin-EDTA, Gibco) and centrifuged at 300 g for 5 min to separate cells from supernatants. Supernatants were centrifugated at 10,000 g for 10 min at 4 °C, aliquoted, and stored at -70/80 °C for secretome analysis. The single cell suspensions were immediately processed for scRNA-seq.

**MDS-L – MSC co-cultures.** The human BM stromal cell line hTERT-MSC (Cat. No. T0523, Applied Biological Materials Inc.) is maintained with aMEM (Gibco) supplemented with 7% human Platelet Lysate (hPL, Macopharma) and Penicillin/Streptomycin (Gibco). The MDS-L cell line (derived from a refractory anemia with ring sideroblasts (RARS) patient and *SF3B1* WT[141]) were kindly gifted by Dr. Kaoru Tohyama (Kawasaki Medical School) and cultured in RPMI 1640 containing Glutamax (Gibco), supplemented with 10% FCS (Gibco), 40 ng/ml GM-CSF (Peprotech), and Penicillin/Streptomycin.

All cell lines were maintained up to 70–80% confluency under standard conditions (37 °C, 5% $CO_2$, > 90% humidity). All cell lines are tested at regular periods for mycoplasma and authenticated by Single Nucleotide Polymorphism (SNP)-profiling (Multiplexion, Heidelberg, Germany).

For secretome analysis, hTERT-MSCs were seeded at a density of 10,000 cells in 96-well Spheroid Microplates (Corning, USA) in 100 μl aMEM + 7% hPL per well and cultured for up to 72 hrs to allow MSC spheroid formation. Afterwards, 1,000 MDS-L cells were added to the MSC spheroids in fresh culture media. MSC spheroid and MDS-L mono-cultures were included as controls. After 96 hrs of co-culture, 50 μl supernatant was collected per well and pooled across wells (up to 200 μl per condition) to collect sufficient material for downstream analyses. Next, the supernatants were centrifuged at 300 g for 5 min, followed by 10,000 g for 5 min to exclude any cell debris, and stored at -20 °C until further use for cytokine analysis.

## Secretome analysis

**Olink (HSPC – MSC co-cultures).** Secretome profiling of the HSPC–MSC co-cultures was performed using a customized Olink® Flex panel with 30 targets (Olink Proteomics AB, Supplementary Data 6). Supernatant was undiluted and incubated with 30 paired oligo-nucleotide-labeled antibodies. After hybridization and DNA polymerization, the reporter sequences were quantified using real-time PCR (BioMark, Fluidigm), and data were processed using the software NPX Manager (Olink Proteomics AB). The resulting threshold cycle (Ct)-data were processed for quality control and normalized, using internal and external controls. The protein levels were reported in absolute values (pg/ml) and normalized as Normalized Protein eXpression (NPX) values. The three technical replicates were merged taking the mean and used for further analysis.

**Luminex (MDS-L – MSC co-cultures).** Cytokine profiling was performed using a custom 20-plex ProcartaPlex kit (Invitrogen, Thermo-Fisher, USA, Supplementary Data 6). Supernatants and standards were processed in duplicate on a MAGPIX instrument (Luminex Corp), following the manufacturer's instructions. The MAGPIX software (Luminex Corp) was used to calculate the cytokine concentrations after the collection of standard curves (pg/ml).

All measured cytokine concentrations were assessed for statistical evaluation and graphical reports were established with Graphpad Prism version 10.4.1 (GraphPad Software, San Diego, California, USA).

## NanoString nCounter gene expression analysis

36 whole BM aspirates (12 MDS patients, 12 CHIP donors, and 12 age-matched Controls), curated as part of the BoHemE study (NCT02867085), were used for NanoString analysis. Total mRNA was isolated using AllPrep DNA/RNA Mini Kit (Qiagen) following the manufacturer's instructions. 150–300 ng RNA was extracted per sample. The expression of 1255 unique genes were analyzed in the NanoString nCounter Pro Analysis System using the PanCancer Immune profiling panel and Immune Exhaustion panel (NanoString Technologies). The *nSolver* software package and nSolver Advanced Analysis module (NanoString Technologies) were used to evaluate the determined transcript counts. Quality-checked raw data was normalized utilizing the geometric mean of the housekeeping reference genes and the code sets' internal positive controls were used to exclude samples that are outliers. Statistics were calculated in R based on normalized and log2-scaled counts applying an empirical Bayes moderated t-statistics tests using limma (v.3.58.1)[142] and corrected for multiple testing adopting the Benjamini-Hochberg procedure. Genes were determined differentially expressed with FDR < 0.05. For comparing CHIP and Control samples, log2-fold changes were ranked based on their magnitude without applying an FDR cut-off and determined as differential with an absolute fold change > 0.5. Gene Ontology (GO) and Kyoto Encyclopedia of Genes and Genomes (KEGG) enrichment of differentially expressed genes was done with clusterProfiler v.4.10.1[143] using all genes measured in the same panel as background. *P* values of GO and KEGG enrichment results were corrected for multiple testing applying

Benjamini-Hochberg procedure and determined significant with an FDR < 0.05. Top 10 significant GO- and KEGG terms were selected by magnitude of FDR for plotting. Additionally, significantly changed genes were submitted to Gene Set Enrichment Analysis (GSEA) against the Hallmark and the C6 gene sets[144,145].

### Healthy BM single cell atlas integration with NanoString data

Publicly available healthy BM scRNA-seq dataset from[47] was acquired. Expression of genes found to be significantly up- or down-regulated between the MDS and Control samples in NanoString analysis (both panels used) at FDR < = 0.05 was aggregated based on either the cell type annotation provided by the authors (Supplementary Fig. 2E) or a simplified annotation grouping related cell types (Fig. 1E) using Seurat's AverageExpression function and visualized using the Complex-Heatmap R package[146].

### Cell isolation/processing of hematopoietic and non-hematopoietic cells for scRNA-seq

BM aspirates from a representative subset of donors in this study (3 age-matched Controls, 3 CHIP, and 4 MDS donors), curated as part of the BoHemE study (NCT02867085), were included for scRNA-seq. The samples were thawed in 100% FCS + 100 μg/ml DNAse I, centrifuged at 300 g for 5 min and suspended in DMEM + 10% FCS + 100 μg/ml DNAse I. When necessary, cell pellets were treated with 1X Red Blood Cell Lysis for 10 min at RT. Cells were washed with 2% FCS/PBS. Live/dead staining (FV780, Thermo Fisher) was performed in PBS, followed by antibody staining for 30 min at 4 °C in CellCover (Anacyte Laboratories) (see Supplementary Data 3 for antibody panel and dilutions). Stained cells were sorted into PBS + 5% BSA with a FACSAria™ Fusion (BD Biosciences) equipped with a 130 μm nozzle into BSA-coated LoBind 1.5 ml tubes (Eppendorf).

### scRNA sequencing
#### 10x Genomics (droplet-based)

**Primary patient samples.** The HSPC and T cell fractions were combined and subjected to scRNA-seq in parallel with the stromal fraction, ensuring clean separation of the populations and minimizing contamination of blood or immune cells in the stromal fractions. 10x Genomics was performed using the Chromium Next GEM (Gel Bead-In Emulsions) Single Cell 3' kit (v3.1, 10x Genomics) according to the manufacturer's instructions. We pooled the T cells (CD45$^{high}$CD34$^-$CD14$^-$CD235a$^-$CD71$^-$CD3$^+$) and HSPC (CD45$^{dim}$CD34$^+$CD14$^-$CD235a$^-$CD71$^-$) fractions in 1 lane and the stromal cells (CD34$^-$CD45$^-$CD14$^-$CD235a$^-$CD71$^-$CD38$^-$) in a separate lane. Sorting and single-cell processing was performed in 3 batches on different days, with each batch consisting of a mix of Control, CHIP, and MDS donors to correct for batch effects. Libraries were prepared based on manufacturer's instructions, with 13–14 PCR cycles for the final library generation. Following, the libraries were equimolarly pooled and sequenced using Illumina NextSeq 2000 with P2 and P3 flow cells.

**HSPC-MSC co-cultures.** The HSPC-MSC co-cultures (including both stromal and HSPC fractions) were subjected to scRNA-seq, using Chromium Next GEM Single Cell 3' kit (v3.1, 10x Genomics), according to manufacturer's instructions. Single-cell processing was performed in three independent batches, each containing one donor per condition (Control, MDS) and a MSC mono-culture control to correct for batch effects. We loaded each condition into a separate 10x lane. Libraries were prepared based on manufacturer's instructions and sequenced using Illumina NovaSeqX plus.

**CEL-Seq2 (plate-based).** The CEL-Seq2 protocol was carried out as according to published information[147]: CD45$^-$CD14$^-$CD235a$^-$CD71$^-$CD38$^-$CD271$^+$ single cells were sorted in 5 384-well plates containing 5 μl of CEL-Seq2 primer solution in mineral oil

(24 bp polyT stretch, a 4 bp random molecular barcode (UMI), a cell-specific barcode, the 5′-Illumina TruSeq small RNA kit adapter, and a T7 promoter), provided by Single Cell Discoveries (Utrecht, the Netherlands). After sorting, the plates were centrifuged for 1 min at 1000 g and immediately placed on ice and stored at −70 °C. Single Cell Discoveries further processed the plates. In brief, ERCC Spike-in RNA Mix (Ambion, 0.02 μl of 1:50,000 dilution) was added to each well before cell lysis with heat shocking. Reverse transcription and second-strand synthesis reagents were dispensed using the Nanodrop II (GC biotech). After generation of cDNA from the original mRNA, all cells from one plate were pooled and the pooled sample was amplified linearly with in vitro transcription[148]. To generate sequencing libraries, RPI-series index primers were used for library PCR. Libraries were sequenced on an Illumina Nextseq500 using 75 bp paired-end sequencing.

### SNP analysis for genotyping of donors in scRNA-seq

**Experimental.** gDNA was isolated from donors BM cells using the Macherey-Nagel NucleoSpin Tissue kit (Fisher Scientific) and was provided in a concentration ranging from 50 - 140 ng/ul for genotyping using the Infinium CoreExome-24 v1.4 BeadChip (Illumina), including 567,218 markers.

**Computational.** Raw intensity data were genotyped using the Illumina Array Analysis Platform Genotyping Command Line Interface v1.1. Resulting GTC files were converted to VCF using Illumina's GTCtoVCF (v1.2.1) software (https://github.com/Illumina/GTCtoVCF) with GRCh37 reference genome and '--skip-indels' option. All VCF files were merged and indexed using bcftools[149].

To increase the number of usable genotypes, we performed imputation using the Michigan Imputation Server[150]. VCF files were split by chromosome and for each of the autosomes imputation was performed using HRC (Version r1.1 2016) reference panel[151], Eagle v2.4 phasing[152], and an Rsq filter threshold of 0.3. Imputed genotypes for individual chromosomes were merged and lifted over to GRCh38 using the LiftoverVcf program from Picard tools by the Broad Institute (https://broadinstitute.github.io/picard/).

### scRNA-seq preprocessing and quality control
#### 10x Genomics data

**Primary patient samples.** Reads were aligned to the GRCh38 (v.2020-A) reference genome and quantified using cellranger count (10x Genomics, v.7.0.0). To assign individual cells to the corresponding donors, we used Souporcell[153] with a list of common variants from 1000 genomes project[154] filtered to include variants with ≥2% allele frequency and a list of known genotypes derived from the SNP analysis. To increase the power of cell assignments, BAM files from individual libraries that share the same donors were combined and provided as input to Souporcell. Counts from cellranger count were adjusted for ambient RNA contamination using SoupX[155]. Corrected counts were analyzed using scDblFinder[156] and used for downstream analysis using Seurat v4.1.1[157].

Cells with less than 200 genes and more than 50% mitochondrial genes per cell as well as the genes found in fewer than 3 cells were excluded from the downstream analysis. After the first round of pre-processing, normalization, and clustering, we identified 2 clusters that were enriched in doublets identified by Souporcell (Supplementary Fig. 4B, C). Additionally, we identified several clusters of hematopoietic cells deriving from libraries of pooled stromal cells, namely erythroid progenitor cells, monocyte progenitor cells, and plasma cells (Supplementary Fig. 3F). All cells were classified as doublets by either Souporcell or scDblFinder, from doublet-enriched clusters, and clusters corresponding to hematopoietic cells deriving from the stromal libraries pools were removed and normalization, clustering and dimensionality reduction was performed again.

**Table 2 | Overview of bone marrow scRNA-seq datasets included in the comparative and integrative analyses**

| Author | Journal | Reference | Condition | Technology |
|---|---|---|---|---|
| Chen et al. | Blood Cancer Discov, Sep 2023 | 83 | Acute Myeloid Leukemia, Healthy | 10x Genomics |
| De Jong et al. | Nat Immunol, June 2021 | 74 | Multiple Myeloma, Healthy | 10x Genomics |
| Bandyopadhyay et al. | Cell, June 2024 | 47 | Healthy | 10x Genomics |
| This study | - | - | CHIP, LR-MDS, Healthy | 10x Genomics, CEL-Seq2 |

**HSPC-MSC co-cultures**. Reads were aligned to the GRCh38 (v.2024-A) reference genome and quantified using cellranger count (10x Genomics, v.9.0.0). Resulting count tables were used for downstream analysis using Seurat v4.1.1[157]. Cells with fewer than 500 and more than 8000 genes expressed as well as those with more than 20% mitochondrial genes per cell were excluded from further analysis.

**CEL-Seq2 data**. CEL-Seq2 data was processed using a custom Snakemake[158] pipeline. In short, a hybrid genome reference and annotation was prepared by combining human genome reference GRCh38 and a gene annotation from Gencode Release 32 with the reference sequences and annotation of the Ambion ERCC RNA Spike-In Mix (Thermo Fisher Scientific). Raw Fastq files were aligned against the hybrid reference using *STAR* aligner (v2.7.10b)[159]. Resulting BAM files were filtered to remove reads without an assigned cell barcode or UMI using SAMtools[149] and GNU grep. Filtered BAM files were used to create the gene count tables using the dropEst pipeline (v0.8.6)[160] with adjusted settings for CEL-Seq2 experiment and counting of any UMIs that overlap either exonic or intronic regions. The results were additionally corrected for UMI collisions and sequencing errors using dropEst. Corrected counts were used for downstream analysis and integrations using Seurat v4.1.1[157]. Cells with fewer than 1000 UMI counts or belonging to a sample with fewer than 10 cells recovered were removed.

**Normalization, dimensionality reduction, and clustering scRNA-seq data**

**Primary BM samples**. Following quality control and prior to dimensionality reduction, the raw counts were normalized using SCTransform v2[161]. 3000 most variable features were selected using SCTransform. Ribosomal, mitochondrial, sex chromosome genes and transcripts were excluded from the variable features. Selected features were used for principal component analysis (PCA)[162] and then uniform manifold approximation and projection (UMAP) on the first 50 principal components[163]. Cells were then grouped into clusters using the Leiden algorithm[164] and the optimal clustering resolution was determined using Clustree[165].

Full dataset was classified into 3 populations of interest: HSPCs, T cells, and stromal cells; and each subset was normalized separately using SCTransfrom as described above. For integration within the HSPC and T cell populations, Seurat objects were split either based on disease status for T cells or the donor ID for HSPCs, renormalized using SCTransform as described above, and integrated using anchor-based workflow from Seurat. Stromal population from 10x was integrated with the CEL-Seq2 data based on the experiment type using anchors integration.

Marker genes of the unsupervised clusters were identified using Seurat's FindAllMarkers function. Genes considered were detected in at least 25% of cells per cluster (min.pct=0.25) and had at least 0.25 log2-fold change relative to all cells outside the cluster (logfc.threshold=0.25, only.pos=T). Differentially expressed genes between clusters were identified using likelihood ratio test for logistic regression model predicting group membership (test = "LR").

**HSPC-MSC co-cultures**. Raw counts were Log-normalized using NormalizeData function. Individual samples were integrated using the Seurat's anchor-based integration. Top 2000 most variable features were identified using FindVariableFeatures function and used to perform the PCA. Cells were clustered using the Leiden algorithm with the resolution parameter of 0.01 resulting in 2 clusters separating HSPC and stromal cells. Individual cells coming from the MSC mono-culture condition, but clustering with HSPCs were removed from further analysis. Average expression of genes corresponding to proteins in the Olink secretome analysis was calculated per sample separately for HSPC and stromal cells.

**Regulon activity quantification in scRNA-seq data**
Transcription factor regulons were inferred using the pySCENIC method (v.0.11.1)[166] via a custom Snakemake pipeline. Raw expression data was randomly split into 10 subsets. Initial co-expression modules were constructed using the GRNBoost2 algorithm and enriched for transcription factor motifs using the human motif databases v10 (hg38_10kbp_up_10kbp_down_full_tx_v10_clust.genes_vs_motifs.rankings.feather and hg38_500bp_up_100bp_down_full_tx_v10_clust.genes_vs_motifs.rankings.feather) from cisTarget (https://resources.aertslab.org/cistarget/databases/) for each of the subsets with 50 repeats each. Resulting regulons were filtered to only include target genes that were identified for a given regulon in at least 5 independent runs and combined. Regulon activities were quantified using the AUCell method of pySCENIC. For the T cells populations, regulons identified in Mathioudaki et al.[65] were used to compute AUCell scores in our scRNA data using the AUCell R package[66].

**Cell type annotation in scRNA-seq data**
Cell clusters from integrated populations (HSPCs, T cells, and stromal cells) were manually annotated based on the expression of marker genes and AUCell activity scores of regulons. For the unfiltered dataset used in SpliceUp predictions, we included HSPCs that came from the sorting for stroma and performed an additional label transfer using the BoneMarrowMap R package[167] with the reference dataset from Roy et al[60]. Cells corresponding to erythroid progenitors were aggregated and added to the "EryPr cluster" that we defined previously.

**Differential gene expression in scRNA-seq between conditions**
Using the annotated cell types within each population, we aggregated all of the raw counts per gene per donor within each cell type. Prior to testing, genes were filtered to include only the ones that had at least 2 fragments per million present in at least half of the donors in either testing condition. We used the DESeq2 R package[168] to test differential expression between conditions with donor's sex as a covariate in the design formula. The resulting estimates for effect sizes were further corrected using the ashr method[169] via DESeq2's function lfcShrink. Resulting log2-fold change values were used to perform Gene Set Enrichment Analysis (GSEA)[145] against the Hallmark gene sets[170] from MSigDB[171] via the clusterProfiler[172] R package. Enriched terms with adjusted *p* value <= 0.05 were used for visualization, unless specified otherwise. For the comparison of iMSCs between CHIP and MDS, we only used data from 10x, because sample size was not big enough to properly account for multiple covariates.

**Cell-cell interactions analysis of scRNA-seq data**
To infer the potential cell-cell interactions in the scRNA-seq data, we used the NICHES method[87]. The fully integrated dataset was split based

on donor identity and for each donor-specific subset the RunNICHES function from the NICHES package was run with Omnipath[173] as a database for ligand-receptor interactions. Individual datasets were merged using the merge.Seurat function of Seurat and filtered to remove cell pairs with fewer than 5 interactions inferred. Interactions specific to cell populations of interest were selected using the Seurat's FindMarkers function.

For visualization of the number of interactions per cell pair in Fig. 6H,I, the inferred number of interactions was divided by the total number of possible cell pair combinations that could come from the corresponding cell types in each donor, i.e., within each donor a product of the numbers of cells belonging to corresponding cell types.

### Single-cell trajectory inference scRNA-seq data

To construct single cell trajectories, we used the Monocle3[88,174,175] v1.3.1 R package. A subset of the integrated HSPC cells was converted to a "cell_data_set" object using the as.cell_data_set function from SeuratWrappers R package v.0.3.1. Dimensionality reduction was performed using Monocle 3's function reduce_dimension with max_components=3, reduction_method = "UMAP"[176], preprocess_method = "PCA". Following, the cells were again clustered with the Leiden algorithm[164] using the cluster_cells function with resolution=0.01. Monocle 3's learn_graph function was used to construct the trajectory graph. Three main trajectories − myeloid, lymphoid, and erythroid − were interactively constructed with Monocle 3's order_cells function.

### Integration of single-cell data with public datasets

Three publicly available BM scRNA-seq datasets[47,74,83] were integrated with the scRNA-seq dataset from this study (see Table 2). Stromal cells were extracted from each dataset based on author-provided annotations using the subset function in Seurat (v5.0.3). The datasets were merged into a single list of Seurat objects and subjected to standard preprocessing, including normalization, variable feature identification, and data scaling using the default values. Principal component analysis (PCA) was initially applied as dimensionality reduction. Data integration was performed using Harmony[177], using the donor, dataset, and sequencing technology (10x Genomics or CEL-Seq2) as integration variables (group.by.vars), and PCA as the reduction method. Further dimensionality reduction was achieved using Uniform Manifold Approximation and Projection (UMAP) based on the 50 Harmony components. The Seurat layers were joined using the JoinLayers function from Seurat and the k nearest neighbors were computed by applying the FindNeighbors function. Clustering of the cells was conducted with Seurat's FindClusters function, using a resolution of 0.2 to identify broad cellular populations.

### Curated inflammatory and HSPC-support signature scores

Gene signatures were manually curated partly based on[85] for the HSPC-support signature and based on[74] and[178] for the inflammatory signature (Supplementary Data 5). To quantify the signature scores, we used the AddModuleScore[179] function from the Seurat package.

### Prediction and analysis of SF3B1-mutated cells

**Prediction of SF3B1-mutant clones with SpliceUp.** To investigate the differences between the SF3B1 wild-type (WT) and mutant clones in MDS samples, we applied the SpliceUp tool (v0.15.2)[89]. SpliceUp leverages known missplicing events associated with SF3B1 mutations in hematological malignancies to quantify the usage of normal and cryptic splice sites and use the relative difference between the two to predict the mutational status of SF3B1 in individual cells. By aggregating the information across multiple gene transcripts across the genome, SpliceUp can overcome the sparsity of scRNA-seq data. In lieu of the ground truth about the mutational status of an individual cell, SpliceUp fits a modified logistic regression model using the sample

status of SF3B1 mutation as the response variable. A threshold of 10% for false-positive rate was applied to control the error rate of the model. To increase the power of downstream analyses, we applied SpliceUp to the subset of our 10x data that still included the erythroid progenitors from libraries sorted for stromal populations (see above under "Cell type annotation"). Classification was performed based on the list of SF3B1-associated 3' alternative splice site and skipped exon missplicing events provided by the authors using mean imputation strategy. Cells were considered for classification if at least 2 missplicing events were covered in the sequencing data.

**Differential gene expression between SF3B1 WT and mutant clones.** To quantify differential gene expression between cells predicted to be $SF3B1^{WT}$ or $SF3B1^{MUT}$, we used aggregated expression of each gene from the subset of erythroid progenitors (Fig. 5D) based on the donor identity and predicted mutational status. We then used the DESeq2 to test for differentially expressed genes between the groups of mutant and WT cells using the donor identity as a covariate. Estimated effect sizes were further corrected using the ashr method[169] and p values adjusted using Benjamini-Hochberg procedure.

### Quantification and statistics

**NanoString analyses.** To examine associations between PanCancer/ImmuneExhaustion PCA scores and clinical parameters of bone marrow samples (Supplementary Fig. 2A), we evaluated linear relationships between the top ten principal components (PC1–PC10) and selected metadata variables (age, variant allele frequency (VAF), and C-reactive protein (CRP)). Pearson's correlation coefficient was used to quantify associations, and statistical significance was determined using a two-sided unpaired Student's t-test.

**(sc)RNA-seq analyses.** To assess enrichment of specific cell types in CHIP and MDS versus Control (Fig. 3A, and Fig. 5A, Fig. 6A), we performed a one-vs-rest comparison by computing the number of cells per cell type against all other types within the same general population across conditions. Fisher's exact test was used to assess statistical significance, with Benjamini−Hochberg correction for multiple testing. Cell populations with adjusted p values < 0.05 were considered significantly enriched.

For cell−cell interaction analysis between stromal subpopulations and HSPCs (Fig. 4G, and Fig. 6F), we quantified the total number of interactions per stromal cell type using NICHES and tested for differences using Fisher's exact test, followed by Benjamini−Hochberg correction.

Differential expression of individual HSPC-support and angiogenesis-support genes across stromal subsets in CHIP and MDS (Fig. 4F, and Fig. 7B) was tested using the two-sided Wilcoxon rank-sum test.

In co-culture scRNA-seq experiments (Fig. 5I), the log-transformed average expression of selected genes was compared between conditions using a two-sided unpaired Student's t-test.

**Flow cytometry analyses.** For the diagnostics flow cytometry, the differences in the abundance of major BM populations were assessed by one-way ANOVA followed by Benjamini−Hochberg correction for multiple testing (Fig. 1F, and Supplementary Fig. 1C). T cell subsets were compared across conditions using one-way ANOVA with post hoc Tukey's test (Supplementary Fig. 1D). Inflammatory MSC populations were analyzed using an one-way ANOVA with Dunnett's test (Fig. 3I) and a two-way ANOVA followed by Benjamini−Hochberg correction (Supplementary Fig. 6C). Further details on all flow cytometry statistical tests are provided in the Source Data.

**Imaging on human and mouse BM tissue.** For imaging of the histopathological stains (Giemsa) and immunofluorescence on human and

mouse FFPE bone tissue sections, the following tests were applied: Welch's *t*-test for CXCL12⁺ MSCs quantification (Fig. 4C); one-way ANOVA with post hoc Tukey's test for human BM cellularity (Fig. 3C), IL-1R1⁺ MSCs (Fig. 3G), GNLY⁺, HSP60⁺, and OASL⁺ T cells (Fig. 6D), human vasculature metrics (Fig. 7D,E), nearest distance of objects to vasculature (Fig. 7F), and mouse BM cellularity (Fig. 7I); one-way ANOVA with FDR (Benjamini–Hochberg) correction for mean sinusoidal and arteriolar vessel area in mouse BM (Fig. 7L, Supplementary Fig. 12F); and two-way ANOVA with post hoc Tukey's test for total vasculature quantification in mouse BM (Fig. 7K, Supplementary Fig. 12E). Full statistical details for imaging data are provided in the Source Data.

**Co-culture secretome analysis (Olink and Luminex).** For secretome measurements in co-culture experiments with primary HSPCs or MDS-L cells and BM MSCs (Fig. 5, Supplementary Fig. 9,10), we applied one-way ANOVA with post hoc Tukey's test to identify significant differences in cytokine and secreted factor levels across conditions. Full details are available in the Source Data.

**Peripheral blood counts in MDS patients.** Differences in peripheral blood parameters across MDS mutational subgroups (*SF3B1*, other splicing factors, *DNMT3A/TET2/ASXL1*) were tested using one-way ANOVA with post hoc Tukey's correction (Supplementary Fig. 1B). Full details are available in the Source Data.

All statistical analyses were performed in R or GraphPad Prism (version 10.4.1). For ANOVA, data were tested for normal distribution (Shapiro-Wilk and Kholmogorov-Smirnov), and for sample sizes (n < 10), QQ plots were generated to guide the selection of appropriate statistical tests.

### Reporting summary
Further information on research design is available in the Nature Portfolio Reporting Summary linked to this article.

## Data availability
Single-cell profiling and NanoString nCounter gene expression data has been newly generated. The scRNA-seq and NanoString data have been deposited in Figshare under the (https://doi.org/10.6084/m9.figshare.27643503). Raw data are available on the Gene Expression Omnibus (GEO): 10x and CEL-Seq2 (https://www.ncbi.nlm.nih.gov/geo/query/acc.cgi?acc=GSE309534), (https://www.ncbi.nlm.nih.gov/geo/query/acc.cgi?acc=GSE309536), and NanoString (https://www.ncbi.nlm.nih.gov/geo/query/acc.cgi?acc=GSE309538), (https://www.ncbi.nlm.nih.gov/geo/query/acc.cgi?acc=GSE309540). Data not deposited on GEO are available upon reasonable request through the corresponding authors. For interactive exploration of our scRNA-seq data we created a Shiny web application available (https://shiny-portal.embl.de/shinyapps/app/11_cellmds). The Shiny app was created using the ShinyCell R package[180]. Source data are provided with this manuscript. Source data are provided with this paper.

## Code availability
Code used for the analyses of Nanostring nCounter gene expression data and single-cell transcriptomic profiling are available at (https://git.embl.org/grp-zaugg/BM_CHIP_MDS).

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

## Acknowledgements

We thank the tissue donors of the BoHemE study for their contribution as well as the CHOICE consortium for providing the biomaterial, especially Katharina Zolden. We thank Pr. François Delhommeau, Dr. Pierre Hirsch, and Frédéric de Vassoigne (Saint Antoine Hospital, Paris) for the healthy donor BM-derived MSCs. We thank the tissue bank of the University Medical Clinic in Mainz for processing the samples, as well as the microscopy core facility of the Institute for Molecular Biology (IMB Mainz) for their technical support. We thank the EMBL's FCCF and GeneCore facilities for technical support for the single-cell experiments and sequencing. We thank the ICM's iGenSeq core facility and Y. Marie and M. Coutelier for single-cell sequencing and demultiplexing. We also thank Single Cell Discoveries for processing the CEL-Seq2 experiments. We thank the TU Dresden Flow Facility and Anna-Lena Baumann, Kristin Möbus, and Claudia Richter for support and data analysis. We thank the DKFZ Genomics and Proteomics Core Facility for the genome-wide SNP analysis and Olink analysis. We thank the EMBL IT for providing the infrastructure and support in performing the data analysis. We thank Jean-Karim Heriche for assistance in creation and hosting of the Shiny web application. We are grateful to members from the Zaugg and Guezguez groups for their valuable input throughout the process. Additionally, we thank the Deplancke group (EPFL) for their insightful discussions. Schematics throughout the manuscript were made with BioRender (biorender.com). This work was supported by the German Cancer Consortium (DKTK) Joint Funding Program (DKTK CHOICE) to B.G., M.S., and U.P.; the José Carreras Leukemia Foundation (DJCLS, 04/R 2018) to B.G.; the European Union (ERC, epiNicheAML, 101044873) to J.B.Z.; M.K. is funded by the EC H2020 MSCA-ITN Project ENHPATHY (grant agreement number 860002) to J.B.Z.; the SNSF (P2ZHP3_199669) and EMBO (538-2021) Postdoctoral Fellowships to K.D.P.; the Deutsche Forschungsgemeinschaft (DFG, Project ID 318346496 – SFB 1292) to M.T.; National Institutes of Health grants R01DK118072, R01 AG069010, and U01AG077925 to J.J.T.; J.J.T. is a Scholar of the Leukemia & Lymphoma Society; National Institutes of Health grant F31DK127573 and The Tufts University Scheer-Tomasso Fund philanthropic gift to L.S.S. Part of this work was supported by Cancer United Research Associating Medicine (CURAMUS), the French National Cancer Institute, Ministry of Health, and ITMO Cancer of Aviesan as part of the 2021–2030 Cancer Control Strategy (INCA-DGOS-Inserm_12560) to C.C. Views and opinions expressed are however those of the authors only and do not necessarily reflect those of the European Union or the European Research Council. Neither the European Union nor the granting authority can be held responsible for them.

## Author contributions

K.D.P., K.W., J.B.Z., and B.G. conceived the project and designed the study. K.D.P. and K.W. performed the single-cell and flow cytometry experiments. K.W. performed the MDS-L co-cultures, Luminex, and imaging experiments. M.K. processed and analyzed the scRNA-seq data, NanoString data, and integrated analysis with other datasets. E.S. analyzed the NanoString data and supported experiments. E.V. integrated and analyzed the publicly available scRNA-seq datasets. K.D.P., M.L., C.C., B.G., and T.J. designed and performed primary HSPC co-cultures and subsequent single-cell experiments. K.D.P. performed the Olink

experiment. G.P. and K.S. supported secretomics analysis. M.K. and P.L.M. developed the SpliceUp algorithm. R.W. performed the Nano-String experiment, and M.S. supervised the analysis. L.S.S. performed the in vivo genetic induction of CHIP in mice and isolated femurs. S.W. and U.O. performed diagnostic flow cytometry, retrieved clinical information, and incorporated patient data analysis. M.W. performed flow cytometry. K.R., M.T., and U.P. provided administrative and core facility support and patient samples. K.D.P., M.K., J.B.Z., and B.G. analyzed and compiled the data. K.D.P., K.W., M.K., J.B.Z., and B.G. wrote the manuscript with input from all co-authors. C.C., E.L., M.S, J.J.T., J.B.Z., and B.G. supervised (certain aspects of) the project.

## Funding

## Competing interests

The authors declare no competing interests.

## Additional information

¹Molecular Systems Biology Unit, European Molecular Biology Laboratory (EMBL), Heidelberg, Germany. ²Department of Internal Medicine III, Hematology and Medical Oncology, University Medical Center of of the Johannes Gutenberg University, Mainz, Germany. ³German Cancer Consortium (DKTK), Partner Site Frankfurt/Mainz, German Cancer Research Center (DKFZ), Heidelberg, Germany. ⁴Laboratory of Systems Biology and Genetics, Institute of Bioengineering, School of Life Sciences, École Polytechnique Fédérale de Lausanne (EPFL), Lausanne, Switzerland. ⁵Department of Biomedicine, University Hospital Basel, University of Basel, Basel, Switzerland. ⁶Institute of Molecular Biology (IMB), Mainz, Germany. ⁷Sorbonne Université, CNRS, INSERM, Development, Adaptation and Ageing, Paris, France. ⁸Sorbonne Université, CNRS, INSERM, Institut de Biologie Paris-Seine (IBPS), Paris, France. ⁹Institute of Immunology, Faculty of Medicine Carl Gustav Carus, TU Dresden, Dresden, Germany. ¹⁰National Center for Tumor Diseases (NCT), Partner Site Dresden, Dresden, Germany. ¹¹DKTK, Partner Site Dresden, DKFZ, Heidelberg, Germany. ¹²Institute of Molecular Medicine I, Proteome Research, University Hospital and Medical Faculty, Heinrich Heine University Düsseldorf, Düsseldorf, Germany. ¹³Molecular Proteomics Laboratory, Biological Medical Research Centre (BMFZ), Heinrich Heine University Düsseldorf, Düsseldorf, Germany. ¹⁴Department of Internal Medicine I, Faculty of Medicine Carl Gustav Carus, TU Dresden, Dresden, Germany. ¹⁵The Jackson Laboratory, Bar Harbor, ME, USA. ¹⁶Department of Medicine Huddinge, Center for Hematology and Regenerative Medicine, Karolinska Institute, Huddinge, Sweden. ¹⁷Cell Biology Unit, University Medical Center of the Johannes Gutenberg University, Mainz, Germany. ¹⁸University Hospital Carl Gustav Carus, TU Dresden, Dresden, Germany. ¹⁹Research Center for Immunotherapy (FZI), University Medical Centre of the Johannes Gutenberg University Mainz, Mainz, Germany. ²⁰These authors contributed equally: Karin D. Prummel, Kevin Woods, Maksim Kholmatov. ✉e-mail: judith.zaugg@unibas.ch; bguezgue@uni-mainz.de

