## [Transparent Peer Review file · Nature Communications]

Inflammatory stromal and T cells mediate human bone marrow niche remodeling in clonal hematopoiesis and myelodysplasia

Corresponding Author: Dr Borhane Guezzou

Version 0:

Reviewer comments:

Reviewer #1

(Remarks to the Author)

Prummel and coauthors investigated how clonal hematopoiesis of indeterminate potential (CHIP) and low risk myelodysplastic syndrome (MDS) remodel the bone marrow niche comparing a number of features with the ones of healthy subjects. Differences in inflammatory remodeling in the bone marrow was found between CHIP and MDS. Importantly the authors identified an inflammatory mesenchymal stroma cell (iMSCs) more prominent in MDS which expanded from CHIP as a continuum. An IFN-responsive T cell population seemed to fuel inflammation in the stroma. This latter is the main finding considering that other studies based on gene expression analysis have been suggested an inflammatory state in the groups analyzed by the authors.

The manuscript includes a compendium of methodologies (single cell RNA seq, flow cytometry, immunofluorescent imaging, in vitro co-culture system) to study the cellular composition and drivers in a cohort of aged control donors, CHIP carries with DNMT3A and TET2 mutations and low risk MDS with similar molecular spectrum.

The study is extremely comprehensive and provides an encyclopedia of numerous factors and differences across the trajectory of CHIP-low risk MDS. Dissecting differences across the spectrum is certainly a hard task. The manuscript is very granular but often lacks focus given the enormous amount of data that are packed. The manuscript will benefit of a clearer structure with a summary re-cap of the actual findings. My comments are below:

-Figure 1C: While the PCA component for the immune panel analysis is clearly distinct in the 3 groups analyzed, the PanCancer immune profiling shows a noise with a not clear distinction with a few controls and CHIP samples mixed with MDS. Can the author elaborate on this?

-Figure 1D and E: The figures will suit better for supplemental material. The resolution is poor and they do not add a clear message to the reader.

-It is unclear how many cells were pooled to generate Figure 2B. The authors had a large cohort but the single cell RNA seq analysis was only conducted in a paucity of them.

-The readout of the Y-axis of panel J and K is missing. Legends to the figures are also not clear for these panels. This applies for other panels too (example Figure 4A and D).

-The manuscript will benefit of a final cartoon summarizing the findings and highlighting distinctions vs similarities.

-It is hard for this reviewer to reconcile a therapeutic approach based on the data considering the amount of interplay between pathways. The idea of the authors should be more direct at the end of the discussion.

-Besides the number of methodologies used, was any post-validation of targets done using real-time PCR? If note, most crucial targets should be confirmed.

(Remarks on code availability)

Code availability was sufficient.

Reviewer #2

(Remarks to the Author)

In their manuscript entitled "Inflammatory Mesenchymal Stromal Cells and IFN-responsive T cells are key mediators of human bone marrow niche remodeling in CHIP and MDS," Prummel et. al. use bulk and single cell RNA sequencing studies performed on primary patient samples from healthy donors and patients with clonal hematopoietic stem cell (HSC) disorders associated with increased risk of leukemic transformation to study how HSCs with somatic mutations influence the bone marrow microenvironment. The study includes bulk RNA seq performed on 35 controls, 17 clonal hematopoiesis of indeterminate potential (CHIP) samples and 32 myelodysplastic syndrome (MDS) samples. The authors identify a bone marrow stromal cell population with an inflammatory signature that they term inflammatory mesenchymal stromal cells (iMDS) that are not present in control samples and expand in CHIP and MDS samples. They also identify T cell populations that appear to contribute to the inflammatory bone marrow microenvironment. The bioinformatic analyses are well described except for a novel approach used to identify cells with splicing factor mutations termed SpliceUp. While these findings are notable and consistent with reports from other groups, further consideration of the contribution of specific mutations and some additional functional experiments may strengthen their findings.

Major concerns:

1. "Control" samples with DNMT3A and TET2 mutations with VAFs between 1 – 2 % were still used as controls. Although these mutations do not meet the definition of CHIP because the VAF is < 2%, did they cluster with the CHIP samples on PCA (Figure 1C), and if so, perhaps they should be removed from the control sample group?
2. While initial studies were performed on bulk bone marrow aspirates, the single cell studies were performed on sorted stromal cells, T cells, and hematopoietic stem and progenitor cells (HSPCs). As these populations were enriched, this raises concerns about the conclusions made about the abundance of these populations. In particular, is the strength of interaction with the iMSC due to the relative abundance/recovery of this population compared to the other stromal cells recovered? This seems to explain the loss of Adipo-CAR2 and increase in iMSC interactions noted in MDS samples.
3. Along those lines, it would be helpful to see the differences in the sorted populations (Figure 2C-G) separated by healthy control, CHIP, and MDS.
4. The CHIP and MDS samples are not sequential samples from the same patients and the single cell studies include samples with mutations thought to have different levels of risk of progression to leukemia when found in CHIP (DNMT3A lower risk than TET2) or MDS (SF3B1 lower risk than other splicing factor mutations). The authors also identify this stromal population in a cohort of predominantly low risk MDS samples as well as AML and myeloma datasets. As the risk of leukemia transformation is lower in low risk MDS and multiple myeloma is a cancer that arises from a mature B cell rather than an HSC, can the authors explain why the iMSC population is so abundant in all of these diseases and why myeloma was used for validation.
5. Does the change from a HSPC nonsupportive to supportive signature have any association with a change in the VAF of the somatic mutation in CHIP or MDS samples?
6. The differences in lineage trajectories in Figure 5 are not clear. Could this be due to including samples with different somatic mutations in the analyses? Can the authors show statistical differences more clearly?
7. More information is needed on the SpliceUp algorithm. How is this method more sensitive? What is the minimum number of events needed to call a cell as mutant vs wild type? Why did the authors choose to focus on erythroid progenitors and how do their differential gene expression and splicing analyses compare to other studies?
8. The functional studies of MDS HSCs and stromal cells are missing a normal control, namely healthy CD34+ cells cultured with the stromal cell line.
9. Their secretome analysis also excludes what the mutant HSCs may be secreting in response to contact with the stroma. Can these data be recovered?
10. The murine studies are also missing aged mutant bone marrow samples as controls. Can different strategies be used to isolate vascular niche cells from human bone marrow samples to strengthen the claims at the end of the study?

Minor concerns:

1. The authors use MSC for multipotent stromal cells rather than mesenchymal stromal cells on page 15. Is this intentional?
2. The immunophenotyping of T cells in Figure S1C seems like a lost opportunity to evaluate whether there are differences in conventional CD4, T reg and CD8 T cells. These data should be included.
3. The authors do not clearly state their recovery percentage from the negative selection approach to isolate stromal cells.
4. Multiple myeloma is not a myeloid malignancy. Their findings are intriguing but raise the issue as to whether patients with MGUS, a lymphoid equivalent of CHIP, would have similar findings in the bone marrow.

(Remarks on code availability)

Version 1:

Reviewer comments:

Reviewer #1

(Remarks to the Author)

This reviewer appreciates that the authors took into consideration the comments and addressed them with clarity.

(Remarks on code availability)

Code availability is appropriate

Reviewer #2

(Remarks to the Author)

Prummel and colleagues have put significant effort into clarifying the already detailed analysis of the interaction between inflammatory mesenchymal stem cells and T cells during myeloid disease progression and include new key experiments in support of their findings. All of my concerns have been thoughtfully addressed in the rebuttal and updated manuscript. Two minor concerns:

1. The heading for the section entitled "inflammatory MSCs are exclusively present in CHIP and MDS" does not prepare the reader for the discussion of this cell population in multiple myeloma. Perhaps it should be changed to "...exclusively present in hematologic malignancies" to stress the fact that they are not present in normal cells.
2. The SpliceUP methods included in this manuscript would not equip a reader to be able to validate the reported findings. Will the draft manuscript be available on BioRxiv give readers access to a full description of this new method?

(Remarks on code availability)

REVIEWER COMMENTS

In response to the reviewer's comments, we have revised the manuscript to incorporate data from additional experiments and data analysis, and have updated several figures accordingly. These changes are highlighted in the revised version. We believe the revisions have significantly improved the clarity and overall strength of the manuscript. We appreciate the reviewers' constructive feedback, which has helped enhance the quality of our work.

Summary of major additions:

- We have extended our co-culture experiments to primary cells from CHIP, MDS, and healthy HSPCs co-cultured with primary MSCs and measured their effect on the secretion of specific cytokines. This confirmed our findings that MDS is not affecting the inflammation of MSCs but rather downregulates niche factors, such as CXCL12. We compared this analysis to a co-culture experiment we performed with the MDS-L cell line and MSCs, on which we performed Luminex for secretome analysis. These 2 co-culture experiments are now presented instead of the original experiment.
- We expanded the diagnostics flow cytometry on the T cell population, by assessing CD4/CD8 T cells and Regulatory T cells (Tregs).
- We prepared a separate manuscript describing *SpliceUp*, our computational tool for inferring *SF3B1*-mutant cells from single-cell transcriptomic data. The manuscript is in submission with another journal. We attached the manuscript draft to this response.

Reviewer #2 (Remarks to the Author):

Prummel and co-authors investigated how clonal hematopoiesis of indeterminate potential (CHIP) and low risk myelodysplastic syndrome (MDS) remodel the bone marrow niche comparing a number of features with the ones of healthy subjects. Differences in inflammatory remodeling in the bone marrow was found between CHIP and MDS. Importantly the authors identified an inflammatory mesenchymal stroma cell (iMSCs) more prominent in MDS which expanded from CHIP as a continuum. An IFN-responsive T cell population seemed to fuel inflammation in the stroma. This latter is the main finding considering that other studies based on gene expression analysis have suggested an inflammatory state in the groups analyzed by the authors. The manuscript includes a compendium of methodologies (single cell RNA seq, flow cytometry, immunofluorescent imaging, in vitro co-culture system) to study the cellular composition and drivers in a cohort of aged control donors, CHIP carries with DNMT3A and TET2 mutations and low risk MDS with similar molecular spectrum.

The study is extremely comprehensive and provides an encyclopedia of numerous factors and differences across the trajectory of CHIP-low risk MDS. Dissecting differences across the spectrum is certainly a hard task. The manuscript is very granular but often lacks focus given the enormous amount of data that are packed. The manuscript will benefit of a clearer structure with a summary re-cap of the actual findings. My comments are below:

We thank the reviewer for their favorable evaluation of our study. We have carefully revised the manuscript to enhance its clarity and organization, incorporating summary recaps at the end of each section. Major changes are highlighted in blue font in the revised manuscript.

- Figure 1C: While the PCA component for the immune panel analysis is clearly distinct in the 3 groups analyzed, the PanCancer immune profiling shows a noise with a not clear distinction with a few controls and CHIP samples mixed with MDS. Can the author elaborate on this?

Response: We would like to clarify that the key difference between the PanCancer and the ImmuneExhaustion NanoString panels lies in the two distinct targeted gene sets. Thus, the primary difference between the two PCAs arises from the specific gene sets included in each panel. The PanCancer panel includes genes broadly associated with various cancer types, rather than specifically tailored to hematopoiesis, myeloid malignancies, and the bone marrow (BM) niche. Investigating the loadings of the top 5 PCs in each PCA analysis revealed that the major contributing genes to the Immune Exhaustion panel are B cell-related genes (such as different IGH genes). These genes are not included on the PanCancer panel and likely explain the less distinct clustering in the PC1 vs. PC2 scatter plot (**Reviewer Figure 1**). This is in line with the reduced (pre/pro-) B cell population in MDS (**Fig. 1F**). In addition, the loadings reveal that the main driver of PC1 and PC2 is *IL8/CXCL8*, indicative of a general inflammatory environment captured by both panels.

Reviewer Figure 1: Visualization of the PC loadings for the ImmuneExhaustion panel (left) and the PanCancer panel (right). The major difference between the panels is the lack of B cell-related genes (such as all the immunoglobulin genes IgGs) in the PanCancer panel that contribute strongly to PC1 and PC2 for the ImmuneExhaustion panel.

Action: We have added the following sentence: *“Although cluster separation appeared slightly more distinct in the Immune Exhaustion panel, both panels consistently captured transcriptional divergence in MDS, reflecting broad remodeling of the BM immune milieu.”*

- Figure 1D and E: The figures will suit better for supplemental material. The resolution is poor and they do not add a clear message to the reader.

Response and action: We agree with the reviewer's suggestion and removed the original rank plots from **Fig. 1** for improved clarity.

- It is unclear how many cells were pooled to generate Figure 2B. The authors had a large cohort but the single cell RNA seq analysis was only conducted in a paucity of them

Response: We thank the reviewer for pointing out the missing information regarding the cell numbers analyzed in **Fig. 2** and the corresponding figure legend. We have now included the number of recovered cells for each sample in the figure. Additional details on the number of analyzed cells per donor and subpopulation are provided in **Supplementary Table 5**. We also clarified in the main text and Methods that scRNA-seq was performed on a subset of donors from our cohort.

Action: We added the recovered cell numbers to **Fig. 2** and the corresponding figure legend. We added the following sentences to the Methods about cell isolation and processing for scRNA-seq: *"BM aspirates from a representative subset of donors in this study (3 age-matched controls, 3 CHIP, and 4 MDS donors), curated as part of the BoHemE study (NCT02867085), were included for scRNA-seq."* We also clarified how many samples were used for the NanoString analysis: *"36 whole BM aspirates (12 MDS patients, 12 CHIP donors, and 12 age-matched controls), curated as part of the BoHemE study (NCT02867085), were used for NanoString analysis."* Finally, we also added the number of cells passing the scRNA-seq pre-filtering to the sections *"scRNA-seq preprocessing and quality control"*.

- The readout of the Y-axis of panel J and K is missing. Legends to the figures are also not clear for these panels. This applies for other panels too (example Figure 4A and D).

Response: We thank the reviewer for pointing out the missing axis labeling in Figures 3 and 4. We have now added the appropriate labels on the affected panels. Additionally, we have revised the figure legends to improve clarity and ensure they are more informative.

Action: We have revised the figure legends and added the Y-axis labels "signature score" to **Fig. 3J** and **3K**, **Fig. 4A** and **4D**, and **Fig. 7A**. We also updated the Y-axis label of **Fig. 7B**.

- The manuscript will benefit of a final cartoon summarizing the findings and highlighting distinctions vs similarities.

Response and action: We agree with the reviewer's suggestion and have included a schematic summarizing our key findings on BM niche remodeling. This illustration highlights both the shared and distinct features between aged healthy controls, age-matched CHIP, and LR-MDS donors. The summary figure has been included as the final panel in **Figure 7 (Fig. 7M)**.

- It is hard for this reviewer to reconcile a therapeutic approach based on the data considering the amount of interplay between pathways. The idea of the authors should be more direct at the end of the discussion.

Response: We agree with the reviewer that our findings will not lead to an immediate therapeutic approach. We have decided to remove this sentence.

Action: We rewrote part of the discussion and removed: "[...] which may offer new therapeutic targets and strategies for managing these conditions in future studies" and replaced with the following statement: "Overall, these findings underscore the role of the microenvironment in early myeloid disease and may contribute to the development of future strategies aimed at intercepting pre-malignant hematopoiesis".

- Besides the number of methodologies used, was any post-validation of targets done using real-time PCR? If not, most crucial targets should be confirmed.

Response: We appreciate the reviewer's attention to this important point. While we agree that real-time PCR (qPCR) can be a valuable tool for validating transcriptomic data, its application in our study is limited due to the cell populations' rarity. Specifically, the most significant transcriptional changes we observed occurred in rare and scarcely represented cell populations, particularly in the inflammatory mesenchymal stromal cells (iMSCs), which were specifically identifiable due to the single-cell resolution of our dataset.

As shown in **Supplementary Fig. 3C** and **Supplementary Table 5**, the abundance of CD271⁺ MSCs in bone marrow aspirates is low, and isolating sufficient numbers (>50–500 cells) for meaningful qPCR validation is currently not technically feasible. While *in vitro* expansion of sorted MSCs could be an option, this could introduce culture-induced artifacts as widely described in literature, and lead to a confounding interpretation.

To address these limitations, we pursued orthogonal validation strategies at the protein level, including:

- BM tissue imaging and BM aspirate flow cytometric validation of the inflammatory MSC phenotype (iMSCs, using CD44/IL1R1/CD271/CD73).
- BM tissue imaging confirmation of T cell subsets (IFN-responsive, HSP, cytotoxic).

Additionally, we reinforced the robustness of our findings through cross-dataset validation, by integrating our results with independent datasets from healthy donors and patients with AML and MM (integration of MSC scRNA-seq datasets: **Fig. 3L-M**), which consistently revealed similar inflammatory transcriptional programs activated in MSCs.

Notably, several studies have demonstrated correlations between scRNA-seq and qPCR data. For example:

- Wu et al. (Nature Methods, 2014) provide a foundational benchmark for validating scRNA-seq using qPCR techniques.
- Everaert et al. (Scientific Reports, 2017) and Vieth et al. (Nat Commun. 2019) systematically compared expression levels across modalities, showing that while some genes may show platform-specific variability, most targets correlate well across methods.

Taken together, we hope we have convinced the reviewer that our target validation strategy, including protein-level confirmation, is a reasonable approach for assessing these rare cell populations identified in our single-cell dataset.

Reviewer #2 (Remarks on code availability):

Code availability was sufficient.

Reviewer #3 (Remarks to the Author):

In their manuscript entitled “Inflammatory Mesenchymal Stromal Cells and IFN-responsive T cells are key mediators of human bone marrow niche remodeling in CHIP and MDS,” Prummel et. al. use bulk and single cell RNA sequencing studies performed on primary patient samples from healthy donors and patients with clonal hematopoietic stem cell (HSC) disorders associated with increased risk of leukemic transformation to study how HSCs with somatic mutations influence the bone marrow microenvironment. The study includes bulk RNA seq performed on 35 controls, 17 clonal hematopoiesis of indeterminate potential (CHIP) samples and 32 myelodysplastic syndrome (MDS) samples. The authors identify a bone marrow stromal cell population with an inflammatory signature that they term inflammatory mesenchymal stromal cells (iMSCs) that are not present in control samples and expand in CHIP and MDS samples. They also identify T cell populations that appear to contribute to the inflammatory bone marrow microenvironment. The bioinformatic analyses are well described except for a novel approach used to identify cells with splicing factor mutations termed SpliceUp. While these findings are notable and consistent with reports from other groups, further consideration of the contribution of specific mutations and some additional functional experiments may strengthen their findings.

We thank the reviewer for their positive evaluation of our study and their insightful feedback. We have now carefully revised the manuscript to incorporate reviewers’ suggestions, additional functional validation, and refined our discussion accordingly. These changes are highlighted in blue font in the revised manuscript.

Major concerns:

1. “Control” samples with DNMT3A and TET2 mutations with VAFs between 1 – 2 % were still used as controls. Although these mutations do not meet the definition of CHIP because the VAF is < 2%, did they cluster with the CHIP samples on PCA (Figure 1C), and if so, perhaps they should be removed from the control sample group?

Response: We agree with the reviewer that it is important to exclude control samples with low VAF values that may already have an effect, albeit being very small. We can confirm that none of the control samples with 1-2% VAF for *DNMT3A* and *TET2* mutations (1-2%) were included in the NanoString and scRNA-seq experiments (see updated **Supplementary Fig. 1A**).

We also revisited our diagnostic flow cytometry data, which included three controls with 1–2% VAF (donors 17, 21, and 28). As shown in the revised figure below, the exclusion of these donors does not alter the statistical significance between the different assessed immune cell populations (**Reviewer Figure 3**). Therefore, despite the presence of mutations with very low VAF, we can conclude that these donors did not influence any of the analyses.

A w/ donor 17/21/28

B w/o donor 17/21/28

Reviewer Figure 3: A) Diagnostics flow cytometry dot plots for NK cells, NKT cells, CD3+ T cells, CD19+ B cells, and CD10/CD19+ B cells as presented in **Fig. 1G** and **Supplementary Fig. 1C**, including controls 17, 21, and 28. B) Dot plots without controls 17, 21, and 28, showing that excluding those donors only influenced the significance of the CD19+ B cells.

Additionally, we annotated the PCAs with the VAF values for each sample and verified that there is no association between VAF and clustering pattern for the Control vs. CHIP samples, confirming that the separation between the 2 Control/CHIP clusters in the PCAs (**Fig. 1C**) is not driven by VAF (**Reviewer Figure 4**). In addition, we assessed other metadata values such as inflammation status (CRP levels), age, bone density, and sex, none of which showed a clear clustering in PCA space.

Reviewer Figure 4: Principal Component Analysis (PCA) plots colored by metadata. Each panel shows PCA projections of the samples included in the NanoString, stratified by different metadata categories: classification (control, CHIP, MDS), age, sex, DXA (bone density), variant allele frequency (VAF), SF3B1 mutation status (K700E, non-K700E, wildtype (wt)), inflammation status (C-reactive protein (CRP)).

Action: We have updated **Supplementary Fig. 1A** and added the following sentence to the manuscript to clarify this point: *“In contrast, Controls with low-frequency mutations (VAF < 2%) showed no association with molecular features (Supplementary Fig. 1A, Supplementary Table 1), underscoring their distinction from CHIP donors.”*

2. While initial studies were performed on bulk bone marrow aspirates, the single cell studies were performed on sorted stromal cells, T cells, and hematopoietic stem and progenitor cells (HSPCs). As these populations were enriched, this raises concerns about the conclusions made about the abundance of these populations. In particular, is the strength of interaction with the iMSC due to the relative abundance/recovery of this population compared to the other stromal

cells recovered? This seems to explain the loss of Adipo-CAR2 and increase in iMSC interactions noted in MDS samples

Response: We thank the reviewer for raising this point, as it allows us to clarify our approach. For the single-cell analyses, we indeed sorted specific individual populations (T cells [CD3⁺], HSPCs [CD34⁺], and stroma cells [negative for blood markers and/or CD271⁺]). Importantly, all differential abundance comparisons were performed within each single population, using identical sorting gates across all groups, ensuring unbiased statistical quantification. Regarding the inference of cell-cell communications, we normalized the number of detected interactions to the total number of possible cell pairings (i.e., $\# \text{interactions A-B} / [\#A \times \#B]$), thereby accounting for the differences in population sizes and cell recovery.

Regarding the T cell interactions with iMSCs, indeed, the absence of iMSCs in healthy individuals and the loss of Adipo-CARs in MDS inherently result in a lack of corresponding inferred interactions for these specific stromal subsets. In our manuscript, we aimed to highlight two key observations:

1. **Comparing iMSCs to other MSCs in MDS:** iMSCs in MDS interact more frequently with T-cells than any other MSC population in MDS.
2. **Comparing iMSCs in CHIP to iMSCs in MDS:** iMSCs in MDS exhibit a broader and more frequent set of predicted interactions with T-cells compared to iMSCs in CHIP, suggesting an enhanced communication hub role in the MDS disease state.

Furthermore, our observations emphasize that T cells retain similar interaction profiles with the predominant stromal compartment (iMSCs in MDS versus Adipo-CARs in Controls), highlighting a shift in niche preference that mirrors stromal remodeling. Overall, these findings support the idea that altered stromal composition shapes T cell–niche crosstalk and may contribute to disease-associated inflammation and immune dysfunction.

Action: We have clarified these points in the revised manuscript to avoid any misunderstanding. *Using NICHES, we investigated potential interactions between stromal populations and T cells. In both Control and CHIP samples, the Adipo-CAR1 and Adipo-CAR2 populations exhibited the most interactions with T cells, respectively, whereas in MDS, iMSCs were the major interacting stromal population (Fig. 6F,G). Furthermore, the T cell interactions in MDS differed quite dramatically from Controls and CHIP: Adipo-CARs interacted with all CD4/CD8 populations in Controls, mostly with Tregs, CD4_{naive/CM}, and CD4 mixed cells in CHIP, while in MDS, their interactions were primarily with IFN-responsive T cells and CD4EM cells (Fig. 6H). Finally, while MDS-iMSCs strongly interacted with IFN-responsive and CD4EM T cells, CHIP-iMSCs showed very little interaction with any T cell population (Fig. 6I), corroborating the observation that the iMSCs in CHIP and MDS represent two distinct functional populations.*

3. Along those lines, it would be helpful to see the differences in the sorted populations (Figure 2C-G) separated by healthy control, CHIP, and MDS

Response: In the current manuscript, we showed the abundance differences by group in the Spider plots in **Figure 3** for Stroma, **Figure 4** for HSPC, and **Figure 6** for T cells. Based on this comment, we realized that an additional direct visualization per group may bring more clarity, and we have now added these figures in the new **Supplementary Figure 4** and copied below as **Reviewer Figure 5**.

Action: We split old Supplementary Figure 3 into two **Supplementary Figures 3 and 4** and added the bar plots (absolute cell numbers and percentages) to **Supplementary Figure 4**.

Reviewer Figure 5: A-C) Spider plots showing the relative abundance of cell types within the stromal, HSPC, and T cell compartments across Control (green), CHIP (purple), and MDS (orange) donors (part of **Figures 3A, 5A, and 6A**). Cell type proportions are plotted on the radius using a probit transformation. Points outlined in black indicate statistically significant differences compared to controls (see Methods). **D)** Bar plots showing both the absolute cell numbers and relative percentages of the indicated cell populations across Control, CHIP, and MDS samples (**Supplementary Figure 4**).

4. The CHIP and MDS samples are not sequential samples from the same patients, and the single-cell studies include samples with mutations thought to have different levels of risk of progression to leukemia when found in CHIP (*DNMT3A* lower risk than *TET2*) or MDS (*SF3B1* lower risk than other splicing factor mutations). The authors also identify this stromal population in a cohort of predominantly low risk MDS samples as well as AML and myeloma datasets. As the risk of leukemia transformation is lower in low risk MDS, and multiple myeloma is a cancer that arises from a mature B cell rather than an HSC, can the authors explain why the iMSC population is so abundant in all of these diseases, and why myeloma was used for validation.

Response: We appreciate the reviewer's insightful question regarding the presence of iMSCs across different hematological malignancies with distinct risk profiles. We chose to compare our iMSCs with those identified in Multiple Myeloma (MM) because it was one of the few data sets that also described iMSCs. Our goal was not to directly validate our findings using the MM data, but rather to assess if iMSCs with similar inflammatory profiles also emerge in distinct hematological diseases. While we cannot fully explain the similarity (it is an observation we made), we offer our interpretation below, and now also clarified this in the revised manuscript.

As noted by the reviewer, mutations such as *DNMT3A*, *TET2*, and *SF3B1* differ in their risk profiles concerning leukemic progression. *DNMT3A* and *SF3B1* mutations are generally considered to confer lower risk compared to *TET2* or other splicing factor mutations (Jaiswal et al., *NEJM* 2014; Desai et al., *Nat Med* 2018). However, in line with our observations of iMSCs, *DNMT3A*-mutant HSPCs have been shown to enhance inflammatory cytokine production in the BM (Kim et al., *J Exp Med* 2021; Hormaechea-Agulla et al., *Cell Stem Cell* 2021; Jakobsen et al., *Cell Stem Cell* 2024), and *SF3B1* mutations have been associated with innate immune dysregulation and inflammatory signaling (Obeng et al., *Cancer Cell* 2016; Shiozawa et al., *Blood* 2018; Pollyea et al., *J Leukoc Biol.* 2021; Choudhary et al., *eLife* 2022). Therefore, while the absolute risk of neoplastic transformation varies, these mutations can initiate microenvironmental alterations towards a pro-inflammatory niche state.

iMSCs have previously been described in AML and MM. Our analysis confirmed that iMSCs exhibit comparable inflammatory signatures in CHIP, MDS, MM, and AML, suggesting that bone marrow niche inflammation may represent a conserved stromal response to chronic hematopoietic stress, irrespective of disease stage, lineage input (myeloid or lymphoid), or transformation risk. This concept is supported by recent studies identifying inflammatory MSC populations in MM (e.g., Ghamlouch et al. *Exp Hematol Oncol* 2025; Cenzano et al. *BioRxiv* 2024), and is consistent with a review by de Jong et al. (*Nat Rev Immunol* 2024), which

highlights stromal parallels between the AML (myeloid-disease) and MM (lymphoid-disease) bone marrow microenvironments.

We further examined the two distinct iMSC clusters identified in our integrated stromal dataset (Fig. 4 / Supplementary Fig. 7): one shared across MM, MDS, CHIP, and AML, and another more specific to MM/AML. Differential gene expression analysis followed by GO term enrichment analysis revealed differences related to cell migration and cilia formation, suggesting these two clusters relate to different types of (i)MSCs, with features potentially linked to MSC differentiation capacity (Tummala et al. *Cell Mol Bioeng.* 2010) (Reviewer Figure 6). However, detailed investigation of these differences is beyond the scope of this study.

Reviewer Figure 6: GO enrichment analysis of the top 100 up- and top 100 down-regulated genes comparing the shared iMSC cluster (MM, MDS, CHIP, AML) and the MM/AML-specific iMSC cluster (all genes $p_{adjusted} < 0.05$, $|\text{Log}_2\text{FC}| > 1$). Terms upregulated in the shared

cluster relate to chemotaxis and cell migration, while terms enriched in the AML/MM-specific cluster relate to ciliogenesis.

Overall, we acknowledge that our study is cross-sectional, and due to the lack of longitudinal sampling, we can only speculate about the temporal dynamics and long-term consequences of this inflammatory state from CHIP to MDS to AML. We have clarified these points in the revised Discussion to better reflect our interpretation and removed any language implying a direct or linear progression between these conditions. Additionally, we reframed the MM/AML comparison as part of a broader cross-disease analysis of niche remodeling, rather than a validation study, and adjusted the Discussion accordingly to avoid overextending conclusions beyond our data.

Action: We have clarified our interpretation of these findings and integrated them into the revised discussion.

5. Does the change from a HSPC non-supportive to supportive signature have any association with a change in the VAF of the somatic mutation in CHIP or MDS samples?

Response: We thank the reviewer for this interesting question. Within the subset of our cohort for which scRNA-seq data were available, we did not observe a clear correlation between the VAF and the stromal support signature across CHIP or MDS samples (**Reviewer Figure 7**). However, we acknowledge that the current sample size is limited, which precludes definitive conclusions. We agree that future studies with larger and longitudinally sampled cohorts will be important to assess whether clonal burden (e.g., VAF) dynamically influences stromal remodeling, in particular in CHIP.

Reviewer Figure 7: Scatterplot showing the relationship between VAF% and the HSPC support gene signature in MSCs across Control, CHIP, and MDS samples. No significant correlation was observed.

Action: As no significant change was detected, we have opted not to include this analysis in the manuscript.

6. The differences in lineage trajectories in Figure 5 are not clear. Could this be due to including samples with different somatic mutations in the analyses? Can the authors show statistical differences more clearly?

Response: We appreciate the reviewer's concerns and recognize that **Fig. 5C** may not have been entirely intuitive. For clarification, **Figure 5** does not compare lineage trajectories between groups. Instead, we inferred pseudotime trajectories for each lineage across all conditions, assigning a pseudotime value to each cell. We then stratified the cells by condition, ordered them by pseudotime for each lineage, and visualized the expression of *IL1B* and *CXCL8* smoothed along each trajectory. This representation highlights expression dynamics across disease conditions rather than direct trajectory comparisons. Regarding statistical significance, we have quantified the confidence intervals for the gene expression, which are depicted in light grey; we have now increased their contrast for improved visibility (**Fig. 5C**).

Action: We have updated Fig. 5C and clarified the figure legend.

B

C

Reviewer Figure 8: updated Fig. 5C.

7. More information is needed on the SpliceUp algorithm. How is this method more sensitive? What is the minimum number of events needed to call a cell as mutant vs wild type? Why did the authors choose to focus on erythroid progenitors and how do their differential gene expression and splicing analyses compare to other studies?

Response: We will address the questions embedded in these comments below:

- **How is this method more sensitive?** We have compared the simple pileup (piling up all the reads of the mutated *SF3B1* transcript) to the *SpliceUp*, which aggregates the signal from dozens of mis-spliced transcripts. We have submitted the detailed algorithm to another journal, Kholmatov *et al.*, and provide the submitted manuscript as supplementary material to the rebuttal letter (see below). We demonstrated increased sensitivity compared to direct identification of mutations at all levels of coverage (see **reviewer Figure 9** - Figure 2d in the *SpliceUp* manuscript).

[REDACTED]

Reviewer Figure 9: SpliceUp discovers more mutant cells compared to simple piling up of reads from the SF3B1 transcript.

- **What is the minimum number of events needed to call a cell as mutant vs wild type?** This is part of the model-fitting process and depends on the sample. SpliceUp fits a modified logistic regression to the mutation-specific mis-splicing events and estimates the False prediction rate based on wild-type and mutant samples (more details in Kholmatov *et al.*). We used an empirical threshold of at least four events for classification; other cells are filtered out and not classified at all.
- **Why did the authors choose to focus on erythroid progenitors?** We initially performed the analysis across all cell types, but only erythroid progenitors had sufficient cell numbers to provide the statistical power needed to call significant differentially expressed genes.
- **How do their differential gene expression and splicing analyses compare to other studies?** The pathways we find enriched in mutated cells are consistent with findings from other studies using experimental technologies to isolate the mutant cells. For example, the enrichment of translation-associated processes has also been reported in Moura *et al.* (*Cancer Res.* 2024) and Cortez-Lopez *et al.* (*Cell Stem Cell* 2023).

Action: We have updated our analysis to the latest version of SpliceUp and revised the corresponding figure panels (**Fig. 5D-F**). Due to coverage constraints, we focused the study on the erythroid progenitors, and the **Methods** section has been updated accordingly. Additionally, we have now submitted a manuscript describing the main algorithm, which will be published in a separate journal.

8. The functional studies of MDS HSCs and stromal cells are missing a normal control, namely healthy CD34⁺ cells cultured with the stromal cell line.

Response: We fully agree with this suggestion and have revised our manuscript accordingly. We replaced the original mass spectrometry-based proteomics data from MDS-L co-culture supernatants with new functional data generated from co-cultures using primary CD34⁺ HSPCs from our study cohort. Specifically, we isolated CD34⁺ HSPCs from Control, CHIP (*DNMT3A*-mut), and MDS (*SF3B1*-mut) patients and co-cultured them for 96 hours with primary BM-derived MSCs from a healthy donor. We included MSC mono-cultures as an additional control. After co-culture, we collected the supernatants and performed secretome profiling using a targeted Olink panel focused on inflammatory and hematopoietic factors, including several key candidates identified through our single-cell transcriptomic analyses. In parallel, we performed scRNA-seq on the Control and MDS HSPC co-cultures to determine the cellular sources of the cytokines within the co-culture system (hematopoietic or stromal compartment).

The most striking observation was the upregulation of *CXCL12* (a critical mediator for HSCs retention), *CCL2*, and *CSF1/CSF2* (M-/GM-CSF) in co-cultures with control and CHIP-derived HSPCs, but not in co-cultures with MDS-derived HSPCs (**Fig. 5G-I**). scRNA-seq of the co-cultures confirmed that *CXCL12* and *CSF1/CSF2* were predominantly expressed in stromal cells. These findings are consistent with our primary patient data that shows an overall loss of *CXCL12* at both gene and protein levels in MDS stromal cells (**Fig. 4B-C, Supplementary Fig. 7C**), further indicating a direct mechanistic link between MDS HSPCs and loss of stromal support in *CXCL12*. This suggests that MDS HSPCs are impaired in maintaining supportive feedback loops with the stromal cells, which is further supported by the observed reduction of the HSC/MPP cluster within the HSPC populations in the scRNA-seq data of primary patient samples (**Fig. 5A**).

Furthermore, consistent with our single-cell observations (**Fig. 5C,E-F; Supplementary Fig. 8A**), MDS HSPCs did not induce inflammatory programs in stromal cells. Although we observed significant changes in cytokine abundance in the co-cultures compared to MSC mono-cultures (including *IL1B*, *CXCL8*, and interferons; **Supplementary Fig. 9B**), subsequent scRNA-seq analysis revealed that these cytokines were almost exclusively expressed by the HSPC fraction (**Supplementary Fig. 9B**). These results suggest that MDS HSPCs primarily disrupt supportive stromal functions without inducing overt inflammatory signaling.

To complement these findings of MSC stroma modulation by MDS HSPCs, we additionally included Luminex-based secretome data from MDS-L co-cultures (now shown in **Supplementary Fig. 10**), which model later disease stages with more differentiated MDS blasts (represented by MDS-L). Unlike the HSPCs, MDS blasts could induce both inflammatory cues (e.g., *IL1A*, *MIP1*, *IL1RA*) and suppress hematopoiesis and angiogenesis-supportive factors (*CXCL12*, *VEGFA*, *HGF*), pointing to a disease-stage-specific modulation of stromal function.

Taken together, our expanded secretome data support the concept that stromal remodeling in MDS is differentially influenced by the nature and the stage of the hematopoietic input: CD34⁺

HSPC primarily impair niche support, whereas blast-like cells further amplify inflammatory signaling, which consequently leads to distinct effects on BM niche function.

Reviewer Figure 10: (H) Quantification of secreted key cytokines post 96 hrs of co-culture. Mean normalized protein expression (NPX) values (3 technical replicates) from Olink across 4 cytokines/growth factors in the culture supernatant across conditions (MSC mono-culture (MSC-only), Control CHIP, MDS). (I) Corresponding gene expression profiling of cytokines shown in (H) within the stromal and HSPC compartments post co-culture. Normalized gene expression (NGX) values for the secreted factors from scRNA-seq of the total recovered stromal and HSPC fractions from MSC mono-culture (MSC-only), and Control and MDS co-cultures. Median with 95% confidence intervals is shown. Statistical significance: n.s. = not significant; * $q < 0.05$; ** $q < 0.01$ (one-way ANOVA, Tukey's test).

Action: We have integrated these new datasets into the Results section and updated **Fig. 5** and **Supplemental Fig. 9** accordingly.

“To functionally investigate how mutated HSPCs influence stromal cells, we established a co-culture system using primary BM-derived MSCs and CD34+ HSPCs isolated from BM aspirates from donors of each group (Control, CHIP, MDS, n=3 per group, all from the study cohort) (Fig. 5G, Supplementary Fig. 9A). Cells were cultured together for 96 hours, and secreted proteins in the culture supernatants were profiled using the Olink platform, focusing on factors related to hematopoietic support and inflammation (Supplementary Table 6, Supplementary Fig. 9B).

After 96 hrs, Control and CHIP HSPCs formed numerous cobblestone areas, which are clusters of HSCs/MPPs, while MDS HSPCs produced very few (Supplementary Fig. 9A). In the secretome, the most striking observation was that CXCL12, along with other hematopoietic support factors such as M-CSF (CSF1), GM-CSF (CSF2), and CCL2, were strongly induced in the co-cultures with Control and CHIP-derived HSPCs but remained at baseline levels in co-cultures with MDS-derived HSPCs (Fig. 5H). scRNA-seq of the co-cultures confirmed that these HSPC-supportive mediators were predominantly expressed by the stroma cells, except for CCL2 which was also ex (Fig. 5I). This is in line with our earlier overall CXCL12 in primary MDS patient samples at both the RNA and protein levels (Fig. 4B,C).

Consistent with our observations in primary BM (Fig. 5C,E-F, Supplementary Fig. 9A), MDS HSPCs did not induce inflammatory programs in stroma cells (Supplementary Fig. 9B), and HSPC intrinsic expression of pro-inflammatory cytokines such as IL1B, CXCL8, and various interferons were even lower in MDS compared to healthy HSPCs (Supplementary Fig. 9B).

To model a more advanced disease stage, we also co-cultured BM-MSCs with the MDS-L cell line, which represents a more differentiated, blast-like MDS population (Supplementary Fig. 10A). Secretome profiling using the Luminex platform, revealed that, in contrast to HSPCs, MDS blasts induced inflammatory factors such as IL1A, MIP1, IL1RA, while also suppressing key hematopoietic and angiogenic-support factors including CXCL12, VEGFA, and HGF (Supplementary Fig. 10B,C). These findings suggest a disease-stage-specific modulation of stromal function.“

9. Their secretome analysis also excludes what the mutant HSCs may be secreting in response to contact with the stroma. Can these data be recovered?

Response: We thank the reviewer for raising this important point. To address it, we have performed scRNA-seq profiling on the primary co-cultures, allowing us to distinguish whether the secreted proteins originate from HSPCs or MSCs, (see above our response to comment 8). However, since HSPCs do not survive without stromal support or exogenous growth factors, we are unable to fully determine whether genes expressed in HSPCs are intrinsic (i.e., pre-existing prior to culture) or induced by signals from the stromal cells.

10. The murine studies are also missing aged mutant bone marrow samples as controls. Can different strategies be used to isolate vascular niche cells from human bone marrow samples to strengthen the claims at the end of the study?

Response to mouse studies: We acknowledge that including *Dnmt3a*-mutant bone marrow samples from aged mice as controls would be ideal. Unfortunately, the *Dnmt3a;Mx-Cre* mouse model consistently develops severe ulcerative dermatitis by approximately one year of age (see survival curve (**Reviewer Fig. 11**)), which is resistant to treatment and necessitates humane euthanasia. This constraint prevents the collection of bone marrow samples from older mutant mice, preventing examination of this model at more advanced ages.

Response to vasculature studies: Secondly, we agree that a more detailed characterization of vascular endothelial cells would be highly informative. We are actively pursuing this line of investigation; however, isolating vascular endothelial cells from cryopreserved human bone marrow (BM) aspirates has presented significant experimental challenges. These cells are both rare and highly sensitive, and in our experience, they are not reliably detectable in cryopreserved samples. As a result, targeted isolation for downstream RNA sequencing has not been feasible. Notably, this limitation is consistent with findings reported in recent single-cell studies of human BM aspirates.

While spatial transcriptomic and targeted methods with a spatial resolution, such as laser capture microdissection, could be a complementary approach to single-cell methods, applying

them to FFPE bone marrow sections presents additional technical challenges, particularly for capturing the transcriptome of these rare and spatially dispersed endothelial cells, and the compromised RNA quality in FFPE bone marrow. Our current efforts to detect endothelial cells in the VisiumHD data remain challenging and require further investigation, but we believe that these protocols open up a new angle for BM endothelium analysis that will be part of our follow-up projects. Currently, no established protocols exist for specifically capturing endothelial cell transcriptomes from FFPE bone marrow sections.

As an alternative approach, we inferred the transcriptional changes in the endothelium by deconvolving bulk transcriptomics signals using the reference healthy BM stromal atlas, which includes some endothelial cell types from freshly processed hip biopsies (**Fig. 1F, Supplementary Fig. 2H**). With this analysis, we observed that some of the inflammatory signals may originate from the endothelium. Given these limitations, we believe that imaging and morphological description of the vasculature, as performed in our study, remains the only approach to study vasculature remodeling in the human BM.

Action: We have now explicitly added transcriptional profiling of the vascular niche to the Discussion section and updated the Methods section accordingly for the *Dnmt3a* mouse model: “Due to the development of ulcerative dermatitis, old *Dnmt3a*^{fl-R878H/+} mice could not be included in the study.”

Reviewer Fig. 11: Survival curve of *Dnmt3a*^{fl-R878H/+} mice. Kaplan–Meier plot illustrating that mice typically require euthanasia by approximately 12 months of age due to treatment-resistant ulcerative dermatitis.

Minor concerns:

1. The authors use MSC for multipotent stromal cells rather than mesenchymal stromal cells on page Is this intentional?

Response and action: We decided to be consistent with the current literature and adhered to the nomenclature, changing the definition to mesenchymal stromal cells.

2. The immunophenotyping of T cells in Figure S1C seems like a lost opportunity to evaluate whether there are differences in conventional CD4, T reg, and CD8 T cells. These data should be included.

Response: We thank the reviewer for this insightful comment. While the initial diagnostic flow cytometry on BM aspirates was performed using predefined clinical panels that did not include sufficient markers to distinguish specific T cell subsets, we fully agree with the importance of resolving T cell composition.

To address this, we conducted additional T cell profiling by flow cytometry on a subset of our BM cohort (n=3 samples per group), quantifying CD4⁺, CD8⁺, and regulatory T cells (CD4⁺/CD25⁺/FoxP3⁺). While total CD4⁺ and CD8⁺ T cell frequencies remained largely unchanged across groups, we observed a trend toward increased Tregs in CHIP and MDS (**Reviewer Figure 12**), which is consistent with our scRNA-seq findings in which Tregs were significantly more abundant in MDS (**Fig. 6A**).

Notably, although bulk T cell proportions were unchanged, scRNA-seq revealed substantial shifts within the composition of CD4⁺/CD8⁺ T cell subsets. Specifically, we identified an expansion of IFN-responsive and cytotoxic CD8⁺ TEMRA/TEFF cells in MDS, and HSP⁺ CD4⁺/CD8⁺ T cells in CHIP and MDS (**Fig. 6A**). These transcriptionally defined states, often present at low frequencies, are challenging to resolve by conventional flow cytometry, underscoring the added value of single-cell transcriptomics in capturing functionally relevant T cell heterogeneity.

Action: We have clarified these points in the revised manuscript and now reference both the additional flow cytometry in revised **Supplementary Fig. 1D**.

Reviewer Figure 12: Frequencies of CD3⁺, CD4⁺, CD8⁺, and regulatory T cells (Tregs) within CD45⁺ population of BM aspirates (post-Ficoll). Statistical significance: n.s. = not significant (one-way ANOVA, Tukey's test).

3. The authors do not clearly state their recovery percentage from the negative selection approach to isolate stromal cells.

Response and action: We thank the reviewer for highlighting the importance of reporting stromal cell recovery rates. We originally showed in **Supplementary Fig. 3C** the percentage of stromal cells within the negative enrichment FACS gate and provided the recovered stromal cell counts per donor in **Supplementary Table 5**. To address this comment more explicitly, we have now updated **Supplementary Fig. 3C** to include both the percentage and absolute counts of CD271⁺ stromal cells as measured by flow cytometry (**Reviewer Fig. 13**) and recovered with scRNA-seq, and have added the corresponding detailed recovery numbers for each donor to **Supplementary Table 5**. This clarifies the efficiency of our stromal cell enrichment and provides greater transparency for readers.

Reviewer Figure 13: Human BM stromal cell recovery. A) FACS gating strategy and CD271⁺ percentage as currently shown in Supplementary Figure 3C. **B)** Bar plot showing the percentage of CD271⁺ cells within the lineage-negative population (CD45⁻/CD14⁻/CD71⁻/CD235a⁻/CD38⁻) as determined by flow cytometry (dark bars), compared to the percentage of stromal cells identified within the sorted lineage-negative fraction in our scRNA-seq data (light bars) across individual donors included in the study. **C)** Bar plot showing the absolute counts of CD271⁺ cells in the lineage-negative population as determined by flow

cytometry (dark bars) and the number of stromal cells recovered in the scRNA-seq dataset (light bars).

4. Multiple myeloma is not a myeloid malignancy. Their findings are intriguing but raise the issue as to whether patients with MGUS, a lymphoid equivalent of CHIP, would have similar findings in the bone marrow.

Response: We thank the reviewer for this important and insightful comment. Indeed, multiple myeloma (MM) is a lymphoid malignancy, and we agree that its inclusion highlights the broader relevance of microenvironmental remodeling across hematologic disease types. Our observation of a shared inflammatory MSC phenotype across CHIP, MDS, AML (myeloid), and MM (lymphoid) raises the possibility that a common MSC subset may be activated in response to inflammatory cues, independent of the hematopoietic lineage affected.

In line with this, we hypothesize that similar inflammatory remodeling may also occur in MGUS, the premalignant precursor to MM, and the lymphoid counterpart of CHIP. Supporting this idea, recent scRNA-seq data from an MGUS/MM mouse model (Ghamlouch et al., *Exp Hematol Oncol* 2025) identified MSC populations with inflammatory features, characterized by elevated cytokine and chemokine expression, reduced bone structural support, and signs of neovascularization. Additionally, Cenzano et al. (BioRxiv 2024.04.24.589777) reported transcriptional alterations in the BM stroma and vasculature of the MGUS/MM patient cohort. While both studies provide compelling evidence of niche remodeling in MGUS, they rely primarily on mouse scRNA-seq and human bulk RNA-seq data. The bulk RNA-seq dataset lacks the resolution needed to identify rare subsets such as iMSCs directly. We consider this an important direction for future studies to include a more comprehensive investigation of MSCs remodeling in MGUS and explore the parallels with CHIP and other myeloid/lymphoid malignancies.

Action: In response to this point, we have clarified the Results section and further emphasized in the Discussion the similarities between inflammatory stromal populations across myeloid and lymphoid blood malignancies.

Cited references

Cenzano, I. et al. Transcriptional remodeling of the stromal and endothelial microenvironment in MGUS to multiple myeloma progression. *BioRxiv* (2024) doi:10.1101/2024.04.24.589777.

Choudhary, G. S. et al. Activation of targetable inflammatory immune signaling is seen in myelodysplastic syndromes with SF3B1 mutations. *eLife* 11, (2022).

Cortés-López, M. et al. Single-cell multi-omics defines the cell-type-specific impact of splicing aberrations in human hematopoietic clonal outgrowths. *Cell Stem Cell* 30, 1262–1281.e8 (2023).

de Jong, M. M. E., Chen, L., Raaijmakers, M. H. G. P. & Cupedo, T. Bone marrow inflammation in haematological malignancies. *Nat. Rev. Immunol.* 24, 543–558 (2024).

Desai, P. et al. Somatic mutations precede acute myeloid leukemia years before diagnosis. *Nat. Med.* 24, 1015–1023 (2018).

Everaert, C. et al. Benchmarking of RNA-sequencing analysis workflows using whole-transcriptome RT-qPCR expression data. *Sci. Rep.* 7, 1559 (2017).

Ghamlouch, H. et al. A proinflammatory response and polarized differentiation of stromal elements characterizes the murine myeloma bone marrow niche. *Exp. Hematol. Oncol.* 14, 22 (2025).

Hormaechea-Agulla, D. et al. Chronic infection drives Dnmt3a-loss-of-function clonal hematopoiesis via IFN γ signaling. *Cell Stem Cell* 28, 1428–1442.e6 (2021).

Jakobsen, N. A. et al. Selective advantage of mutant stem cells in human clonal hematopoiesis is associated with attenuated response to inflammation and aging. *Cell Stem Cell* 31, 1127–1144.e17 (2024).

Jaiswal, S. et al. Age-related clonal hematopoiesis associated with adverse outcomes. *N. Engl. J. Med.* 371, 2488–2498 (2014).

Kim, P. G. et al. Dnmt3a-mutated clonal hematopoiesis promotes osteoporosis. *J. Exp. Med.* 218, (2021).

Moura, P. L. et al. Erythroid Differentiation Enhances RNA Mis-Splicing in SF3B1-Mutant Myelodysplastic Syndromes with Ring Sideroblasts. *Cancer Res.* 84, 211–225 (2024).

Obeng, E. A. et al. Physiologic expression of sf3b1(k700e) causes impaired erythropoiesis, aberrant splicing, and sensitivity to therapeutic spliceosome modulation. *Cancer Cell* 30, 404–417 (2016).

Pollyea, D. A. et al. MDS-associated SF3B1 mutations enhance proinflammatory gene expression in patient blast cells. *J. Leukoc. Biol.* 110, 197–205 (2021).

Shiozawa, Y., Malcovati, L., Galli, A., et al. Gene expression and risk of leukemic transformation in myelodysplasia. *Blood* 132, 869–875 (2018).

Tummala, P., Arnsdorf, E. J. & Jacobs, C. R. The role of primary cilia in mesenchymal stem cell differentiation: A pivotal switch in guiding lineage commitment. *Cell. Mol. Bioeng.* 3, 207–212 (2010).

Vieth, B., Parekh, S., Ziegenhain, C., Enard, W. & Hellmann, I. A systematic evaluation of single cell RNA-seq analysis pipelines. *Nat. Commun.* 10, 4667 (2019).

Wu, A. R. et al. Quantitative assessment of single-cell RNA-sequencing methods. *Nat. Methods* 11, 41–46 (2014).

REVIEWERS' COMMENTS

Reviewer #1 (Remarks to the Author): This reviewer appreciates that the authors took into consideration the comments and addressed them with clarity.

Reviewer #1 (Remarks on code availability): Code availability is appropriate

Response: *We thank the reviewer for their positive evaluation of our revisions and for acknowledging the clarity of our responses. We are also pleased that our code availability statement is considered appropriate. We believe these revisions strengthen the transparency and reproducibility of the study, and we are grateful for the reviewer's constructive input throughout the review process.*

Reviewer #2 (Remarks to the Author):

Prummel and colleagues have put significant effort into clarifying the already detailed analysis of the interaction between inflammatory mesenchymal stem cells and T cells during myeloid disease progression and include new key experiments in support of their findings. All of my concerns have been thoughtfully addressed in the rebuttal and updated manuscript.

Response: *We thank the reviewer for their positive assessment and are pleased that our revisions and additional experiments have fully addressed the previous concerns.*

Two minor concerns:

1. The heading for the section entitled "inflammatory MSCs are exclusively present in CHIP and MDS" does not prepare the reader for the discussion of this cell population in multiple myeloma. Perhaps it should be changed to "...exclusively present in hematologic malignancies" to stress the fact that they are not present in normal cells.

Response: *We thank the reviewer for this suggestion. We agree that the previous heading was too restrictive. We have revised the section heading to: "Inflammatory MSCs are exclusively present in hematologic malignancies" (page 5, line 246), which more accurately reflects the results and accompanying discussion.*

2. The SpliceUP methods included in this manuscript would not equip a reader to be able to validate the reported findings. Will the draft manuscript be available on BioRxiv give readers access to a full description of this new method?

Response: *We appreciate this important point. To ensure transparency and reproducibility, we have taken the following steps:*

- We have deposited the full methodological description of SpliceUP together with benchmarking analyses as a separate preprint on bioRxiv (DOI: <https://doi.org/10.1101/2025.09.22.677806>).

- In the revised Methods section (page 8-9, lines 995–1018), we now clearly state that the preprint provides a comprehensive description of the algorithm, validation datasets, and user guidelines.

- In addition, all code and example datasets have been made openly available through our GitHub repository (URL provided in the Data & Code Availability section).

These measures will allow readers to fully validate and apply SpliceUP in their own datasets.